



# Full-scale spectra of 15-year time series of near-surface horizontal wind speed on the north slope of Mt. Everest

Cunbo Han[1,3], Yaoming Ma[1,2,3,4,6], Weiqiang Ma[1,3], Fanglin Sun[5], Yunshuai Zhang[1,3], Wei Hu[1,3], Hanying Xu[1,3,4], Chunhui Duan[1,3,4], Zhenhua Xi[1,3]

[1]Land-Atmosphere Interaction and its Climatic Effects Group, State Key Laboratory of Tibetan Plateau Earth System, Resources and Environment (TPESRE), Institute of Tibetan Plateau Research, Chinese Academy of Sciences, Beijing 100101, China.
[2]College of Hydraulic & Environmental Engineering, China Three Gorges University, Yichang 443002, China.
[3]National Observation and Research Station for Qomolongma Special Atmospheric Processes and Environmental Changes,
Dingri 858200, China.
[4]University of Chinese Academy of Sciences, Beijing, 100049, China.
[5]Key Laboratory of Land Surface Process and Climate Change in Cold and Arid Regions, Northwest Institute of Eco-Environment and Resources, Chinese Academy of Sciences, Lanzhou 730000, China
[6]College of Atmospheric Sciences, Lanzhou University, Lanzhou 730000, China

*Correspondence to*: Cunbo Han (cunbo.han@hotmail.com) and Yaoming Ma (ymma@itpcas.ac.cn)

**Abstract.** Wind speed spectral analysis is of great importance for understanding boundary-layer turbulence characteristics, developing atmospheric numerical model, and assessing wind energy. 15-year time series of near-surface horizontal wind data from the national Observation and Research Station for Qomolongma Special Atmospheric Processes and Environmental Changes (QOMS) on the north slope of Mt. Everest has been used to investigate the full-scale wind spectrum
in the frequency range from about 10 $yr^{-1}$ to 5 Hz. The annual average wind speed showed almost no detectable trend from 2006 to 2018 at the QOMS station. Three peaks were identified in the full-scale spectra at the frequencies of 1 $yr^{-1}$, 1 $day^{-1}$, and 12 $hr^{-1}$, respectively. The 12 $hr^{-1}$ peak is evident in spring and summer but disappears in winter, indicating the seasonal differences in local circulations at the QOMS station. The spectral density was the highest on the low-frequency side of the diurnal peak and in the microscale frequency range ($f \geq 1 \times 10^{-3}$ Hz) in winter, indicating frequent synoptic weather events
and vigorous turbulent intensity generated by shear due to strong wind during winter. An obvious spectral gap around the frequency of $4.5 \times 10^{-4}$ Hz was observed in the composite seasonal and daily spectrum in winter, while the spectral gap disappeared in summer. The combination of low spectral density in the mesoscale frequency range, high spectral density in the microscale frequency range, and strong turbulence intensity contributes to the spectral gap in winter. Wind speed spectral analysis across different scales provides valuable input for numerical evaluations of wind fields and valley wind circulations
in the mountainous region.





## 1 Introduction

Mt. Everest (Qomolangma), the highest mountain on Earth, is located in the central part of the Himalayas on the southern edge of the Tibetan Plateau (TP). The Mt. Everest region is at the forefront of global climate change and exbits unique responses to climate change different from other places of the TP (Kang et al., 2022; Ma et al., 2023). Over the past decades, it has experienced significant warming, substantial decrease in precipitation, glacier and snow retreat, and permafrost degradation (Yang et al., 2014; Salerno et al., 2015; Xu et al., 2016; Han et al., 2021; Kang et al., 2022; Ma et al., 2023). The changes affect local and regional atmospheric circulations, especially the local land-atmosphere interactions. To monitor the atmospheric and environmental conditions, a comprehensive observation network has been established in the north slope of Mt. Everest since 2005 (Ma et al., 2023). Based on the observation data, land surface processes, atmospheric boundary processes, atmospheric aerosols and pollutants have been intensively studied (Chen et al., 2013; Han et al., 2015; Sun et al., 2018; Lai et al., 2021; Kang et al., 2022; Ma et al., 2023).

Wind speed spectra, which describe the frequency distribution of wind speed energy within the atmosphere, are of great importance to assess whether the wind speed energy is evenly distributed over different frequencies or concentrated within a certain frequency range. The spectra are fundamental in understanding the characteristics of atmospheric flow, turbulent transport processes, and wind energy distribution (Van der Hoven, 1957; Fiedler and Panofsky, 1970; Larsén et al., 2013; Kang and Won, 2016; Larsén et al., 2016; Gyatso et al., 2023). In practice, wind spectral analysis is widely used in understanding the dynamics of atmospheric boundary-layers (Zou et al., 2020; Larsén et al., 2021; Li et al., 2021; Lin et al., 2021; Williams and Qiu, 2022), validating atmospheric numerical models (Dipankar et al., 2015; Schalkwijk et al., 2015), and assessing wind energy resource (Escalante Soberanis and Mérida, 2015; Watson, 2019; Effenberger et al., 2024; Liu et al., 2024).

There is a long history of studies on the spectral characteristics of atmospheric boundary-layer horizontal wind velocity. Van der Hoven (1957) analyzed a full-scale spectrum of horizontal wind speeds measured at 91 m, 108 m, and 125 m heights at the Brookhaven National Laboratory. The wind speed spectrum has two majority peaks: one located at a period of 4 days and a second peak occurring at a period of about 1 minute, corresponding to the synoptic-scale and turbulent motions, respectively. Between the two major peaks is a wide spectral gap at a frequency ranging from 2 h to 10 min due to lack of physical processes that could support wind speed fluctuations in the frequency range. This is the so-called "gap" at a period of about 1 hr separating the three-dimensional (3D) microscale turbulence from the two-dimensional (2D) mesoscale to macroscale motions (Stull, 1988).



There are a lot of discussions on the existence of the spectral gap. Many studies observed a local minimum in the wind speed spectrum at periods of about 1 h and confirmed the gap existence (Fiedler and Panofsky, 1970; Smedman-Högström and Högström, 1975; Gomes and Vickery, 1977; Kaimal and Finnigan, 1994; Yahaya et al., 2003; Kang and Won, 2016; Larsén et al., 2016; Li et al., 2021), although the gap was not as significant as that of Van der Hoven (1957). However, the spectral gap is not always well defined. Based on the assumption of the spectral gap, we might expect there to be little variability in the wind speed on these time scales, and that in certain atmospheric conditions the gap may not exist at all (LeMone, 1976; Gjerstad et al., 1995; Heggem et al., 1998). Studies question the existence of the gap in Van der Hoven (1957)'s wind speed spectrum mainly because the high-frequency region was observed during the passage of a hurricane, which increases the spectral density in the high-frequency range compared to standard atmospheric conditions (Smedman-Högström and Högström, 1975; Kang and Won, 2016; Larsén et al., 2016). Multiple atmospheric processes are reported that contribute to the generation of variations in the spectral gap region, including large cumulus clouds (Stull, 1988), convective cells (Gjerstad et al., 1995; Heggem et al., 1998), and horizontal roll vortices (Heggem et al., 1998). Larsén et al. (2016) suggested that the gap is jointly regulated by the variation in horizontal wind variations from the two-dimensional mesoscale motions and three-dimensional boundary-layer turbulence, and that it may be visible or invisible depending on the relative contributions of microscale and mesoscale motions to the fluctuations. Moreover, the presence or absence of a spectral gap also depends on the sampling strategy. In the absence of rolls, the spectral gap would be evident and sufficiently large if the data are collected from a fixed location and only along the wind direction. However, if the data is measured perpendicular to the roll vortices, for example, in an aircraft flying normal to the rolls, a single spectral peak indicating the combination of rolls and 3D large convective eddies will be obtained, and the spectral gap will vanish.

The characteristics of the wind speed spectrum are influenced by various factors, such as topography, land surface conditions, environment, and observations. Therefore, wind speed spectra at different locations and heights have been intensively investigated and compared (Wieringa, 1989; Högström et al., 2002; Larsén et al., 2013; Kang and Won, 2016; Larsén et al., 2016; Li et al., 2021; Lin et al., 2021; Effenberger et al., 2024). Larsén et al. (2013) analyzed the spectral structure of mesoscale winds at two offshore sites and found that the climatological wind spectra show universal characteristics consistent with findings in the literature. Kang and Won (2016) investigated the spectra of 5-year wind speed time series at heights of 10 m and 100 m, observed at the Boulder Atmospheric Observatory (BAO) tower on the eastern slope of the Rocky Mountains in Colorado, about 25 km east of the foothills. They suggested that the spectral gap at periods of about 1 h is not pronounced, while an obvious diurnal peak exists, which is often absent or less noticeable in the spectra at coastal sites. Lin et al. (2021) analyzed the wind spectra observed at 8 urban surface meteorological weather stations in Hong Kong and found that spectra peaks at 1 year, 1 day, and 12 hour are observed at all stations, and the spectral characteristics vary with topography and seasons. Li et al. (2021) studied full-scale spectra of multilayer wind time series measured on a 325 m high



urban meteorological tower in the northern center of Beijing. The results show that the spectral gap exists in the lowest two

layers, becomes less pronounced, and eventually disappears with increasing height. Moreover, the 1 year and diurnal peaks were observed in all layers. The 12-hour spectral peak observed in Beijing is probably related to the diurnal cycle of upslope and downslope winds, as there are mountains to its west and north. The 12-hour cycle in near-surface wind is also frequently observed over tropical and subtropical oceans. Semidiurnal oscillations in both eastward and northward winds were documented by Lindzen and Chapman (1969) using data from a subtropical island site (Terceira, Azores) in the Atlantic.

Jacobs (1980) reported semidiurnal peaks in surface wind speed from ship observations during the GATE experiment. More recently, Ueyama and Deser (2008), using buoy data (1993–2004) from the Tropical Atmosphere Ocean array, found that in the tropical Pacific Ocean the amplitude of semidiurnal variation in the zonal wind component is roughly twice that of the diurnal variation. The 12-hour cycle of near-surface wind speed over the ocean is generally attributed to atmospheric tides, with its amplitude being particularly pronounced under clear-sky conditions (Lindzen and Chapman, 1969; Ueyama and

Deser, 2008; Hsu et al., 2025). Therefore, issues such as the existence of the spectral gap and the 12-hour spectral peak are still being discussed.

It has long been known that horizontal temperature gradient induced by complex terrain may generate well-defined thermal circulations with distinct diurnal cycle (Whiteman, 2000). A typical mountain-valley circulation system that winds blow up

the terrain (upslope and up-valley) during daytime and blow down the terrain (downslope and down-valley) during nighttime. Similar mountain-valley winds circulations are observed in the Himalayas, for example, in the Kali Gandaki Valley in Nepal Himalayas (Egger et al., 2000). However, it is frequently observed that strong up-valley winds persist from noon until sunset and weaker up-valley winds during nighttime, while weak down-valley winds below from sunrise to noon time at a site in the Rongbuk Valley 30 km north of the peak of Mt. Everest (Sun et al., 2007; Sun et al., 2018). Strong down-valley winds

are thought to be caused by glacier winds (Sun et al., 2007). However, simulation results indicate that glacier winds do not extend that far and are counteracted by daytime up-valley winds (Cai et al., 2007). Recently, Sun et al. (2018) argued that the temperature gradient between the north and south slopes of the Himalayas allows up-valley winds from the southern slopes to enter the TP, resulting in strong down-valley winds in the northern slopes of the Himalayas. Therefore, the formation processes of the strong noon-to-sunset down-valley winds are still debated, and its characteristics and driving mechanisms of

formation need more in-depth studies.

This study aims to advance the knowledge of the full-scale horizontal wind speed spectrum in mountainous regions by using extensive datasets of 10-min horizontal wind data and 3D sonic anemometer wind data observed at the northern slope of Mt. Everest over the past 15 years, from September 7, 2005, to June 12, 2019. In addition, it is also a better understanding of the

characteristics of the local circulations on the north slope of Mt. Everest. Section 2 gives the measurements and wind data, as



well as the methodology by which wind data is processed, and the wind spectra are calculated and analyzed. Section 3 presents results and discussion, followed by summary and conclusions in section 4.

## 2 Data and methodology

### 2.1 The QOMS observatory

**Figure 1: Instruments (the PBL tower and the EC tower) and elevation map around the QOMS station (from Google Earth). Black lines are isolines of elevation. South is indicated on the upper-right corner of the figure.**

The national Observation and Research Station for Qomolongma Special Atmospheric Processes and Environmental Changes (QOMS) was established in 2005 and has been gradually developed into a comprehensive observation network



including six observation stations in total on the northern slope of Mt. Everest (Ma et al., 2023). The scientific objective of the QOMS is to advance the understanding of land-atmosphere interactions and related climate impacts on the northern slope of Mt. Everest. The QOMS station is situated in the Rongbuk Valley at an elevation of 4276 m above sea level, approximately 30 km north of the summit of Mt. Everest (Figure 1). The station lies at the bottom of the Rongbuk Valley,

which runs in a northeast-southwest direction (approximately 10° to 190°), with a width of about 1.5 kilometers at the station. The ground surface around QOMS is flat and is covered with sand, gravel, and sparse short grass. A planetary boundary-layer (PBL) tower (Vaisala, MILOS520), an eddy-covariance (EC) tower (Campbell CSAT3 3D sonic anemometer and Li-Cor Li-7500 infrared gas analyzer), and a four components radiation budget system (Kipp & Zonen, CNR-1) have operated since September 2005 and are maintained regularly. The PBL tower is equipped with five levels of air temperature, specific

humidity at heights of 1.5, 2, 4, 10, and 20 m, three levels of wind direction at heights of 1.5, 10, and 20 m, and six layers soil temperature and moisture observations buried at depths of 5, 10, 20, 40, 80, 160 cm. The eddy covariance system is mounted at the height of 3.5 m. Many more instruments have been set up at the QOMS station since 2005, especially after 2019, supported by the second Tibetan Plateau Scientific Expedition and Research Program. More details on the instruments at the QOMS station are found in Ma et al. (2023).

**2.2 Wind data**

We analyzed two types of wind data in this study, 10-min horizontal wind velocity and direction measured with combined cup anemometers and wind vanes mounted on the PBL tower and 10-Hz 3D wind data observed by a 3D sonic anemometer installed on the eddy-covariance tower. The PBL tower and eddy-covariance system have been in operation since September 2005. However, 10-min wind data observed by the PBL tower are not available between June 2019 and December 2019.

Thus, 10-min wind data from September 7, 2005 to June 12, 2019 are analyzed in this study. The 10-m wind data availability is summarized in Table 1 for each year and at the five levels over the 15-year study period. The yearly data coverage of 10-min wind data ranges from 90.1 to 100%, and is generally higher than 95% except for 2008, 2010, and 2013. The 10-Hz sonic wind data were observed at a height of 3.5 m. Data are available from 2005 to 2019. We analyzed the data-available days for each year and found that the data availability was highest in 2015 and 2016. Thus, the 10-Hz sonic wind data from

2015 and 2016 were used to analyze the wind spectra from about $10^{-5}$ to 5 Hz. The 10-Hz 3D sonic wind data availability of each month in 2015 and 2016 is summarized in Table 1 as well. Only days without missing data were selected. All the data used in this study have been published in Ma et al. (2020) and are publicly accessible at the National Tibetan Plateau Data Center (https://doi.org/10.11888/Meteoro.tpdc.270910, last access on June 12, 2024).






**Table 1: Yearly 10-min wind data availability at the five levels on the PBL tower at the QOMS station over the 15-year study period from September 7, 2005 to June 12, 2019, where for each day the data coverage is greater than 99.0%. Available days in each month of the 10-Hz sonic wind data availability at the QOMS station in 2015 and 2016, and for each day the data coverage is 100%.**

| Freq | h (m) | 2005 | 2006 | 2007 | 2008 | 2009 | 2010 | 2011 | 2012 | 2013 | 2014 | 2015 | 2016 | 2017 | 2018 | 2019 |
|------|-------|------|------|------|------|------|------|------|------|------|------|------|------|------|------|------|
| 10-min | 1.5 | 99.7 | 99.5 | 99.7 | 90.1 | 98.2 | 54.3 | 99.2 | 99.5 | 90.2 | 95.9 | 100 | 100 | 97.7 | 96.7 | 97.8 |
| | 2 | 99.7 | 99.5 | 99.7 | 90.1 | 98.2 | 91.8 | 99.2 | 99.5 | 90.1 | 95.9 | 100 | 100 | 97.7 | 96.7 | 97.8 |
| | 4 | 99.5 | 99.5 | 99.7 | 90.1 | 98.2 | 91.8 | 99.2 | 99.5 | 90.1 | 95.9 | 100 | 100 | 97.7 | 96.7 | 97.8 |
| | 10 | 99.7 | 99.5 | 99.7 | 90.1 | 98.2 | 91.8 | 99.2 | 99.5 | 90.1 | 95.9 | 100 | 100 | 97.7 | 96.7 | 97.8 |
| | 20 | 99.7 | 99.5 | 99.7 | 90.1 | 98.2 | 44.0 | 0.0 | 6.9 | 90.1 | 95.9 | 100 | 100 | 97.7 | 96.7 | 97.8 |
| **Freq** | **Year** | | **Jan** | **Feb** | **Mar** | **Apr** | **May** | **Jun** | **Jul** | **Aug** | **Sep** | **Oct** | **Nov** | **Dec** | **Total** | |
| 10-Hz | 2015 | | 29 | 28 | 29 | 29 | 30 | 30 | 29 | 30 | 29 | 30 | 29 | 30 | 352 | |
| | 2016 | | 30 | 29 | 30 | 29 | 30 | 29 | 30 | 30 | 29 | 30 | 29 | 30 | 355 | |






## 2.3 Method of calculating wind speed spectrum

The power spectra are calculated using the Fourier transform method, and a linear detrending is applied to the wind speed data time series. As the Fourier transform requires a continuous time series, thus the missing data points in the time series

must be filled. Kang and Won (2016) tested the sensitivity of spectra calculation to the filling methods, including linear, nearest-neighboring, zero-, first-, second-, and third-order spline interpolation methods, and pointed out that the spectra results are insensitive to the interpolation method. Thus, in this study, the few missing data in the 15-year 10-min wind time series were filled in using the linear interpolation method with data before and after the gaps. For the 10-Hz sonic wind data, only days without missing data points were chosen to calculate spectra for high frequencies.


The power spectral densities are not evenly distributed on a logarithmic scale, sparse in the low-frequency end and dense in the high-frequency end. Frequency smoothing of the power spectra densities is commonly used to extract a representative spectral curve from the estimates (Kaimal and Finnigan, 1994). A frequency smoothing method, in which the averaging interval keeps expanding with frequency, is applied in this study. The number of estimates in each nonoverlapping block

increases exponentially as a function of frequency to yield about seven to eight estimates per decade. In practice, the first few estimates are accepted as they are, then the number of estimates is increased in steps 3, 5, 7, and so on until the density of smoothed estimates per decade reaches seven or eight. Details can be found in section 7.4 of Kaimal and Finnigan (1994).

## 3 Results and discussion

### 3.1 Horizontal wind at QOMS

Wind roses calculated from the 10-min wind time series at 1.5, 10, and 20 m heights for the 15-year period are shown in Figure 2. The compass roses with 16 wind directions show that about 71.5% of the winds follow the direction of the river valley. Specifically, at the height of 1.5 m, northerly to northeasterly winds account for 28.7%, while southeasterly to southwesterly winds make up 42.8%. The wind velocity is mainly below 4 m/s. The predominant wind directions are almost unchanged from 1.5 to 10, and 20 m, north-northeast and south winds are still the prevailing wind directions. While the wind

speed increased significantly with height, a large portion of wind speed is higher than 10 m/s, especially for the westerly and southerly winds.





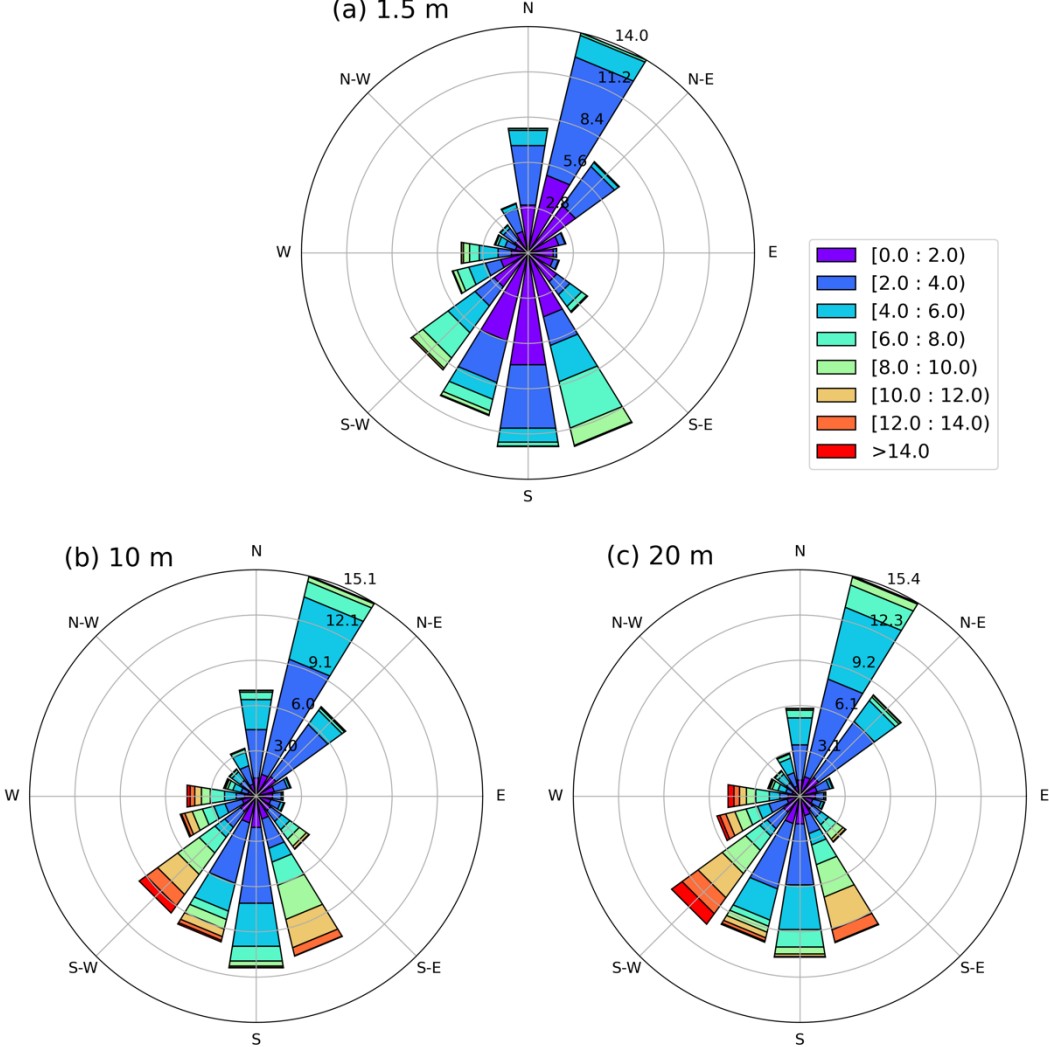

**Figure 2: Wind roses calculated from the 15-year 10-min wind data at (a) 1.5, (b) 10, and (c) 20 m at the QOMS station.**





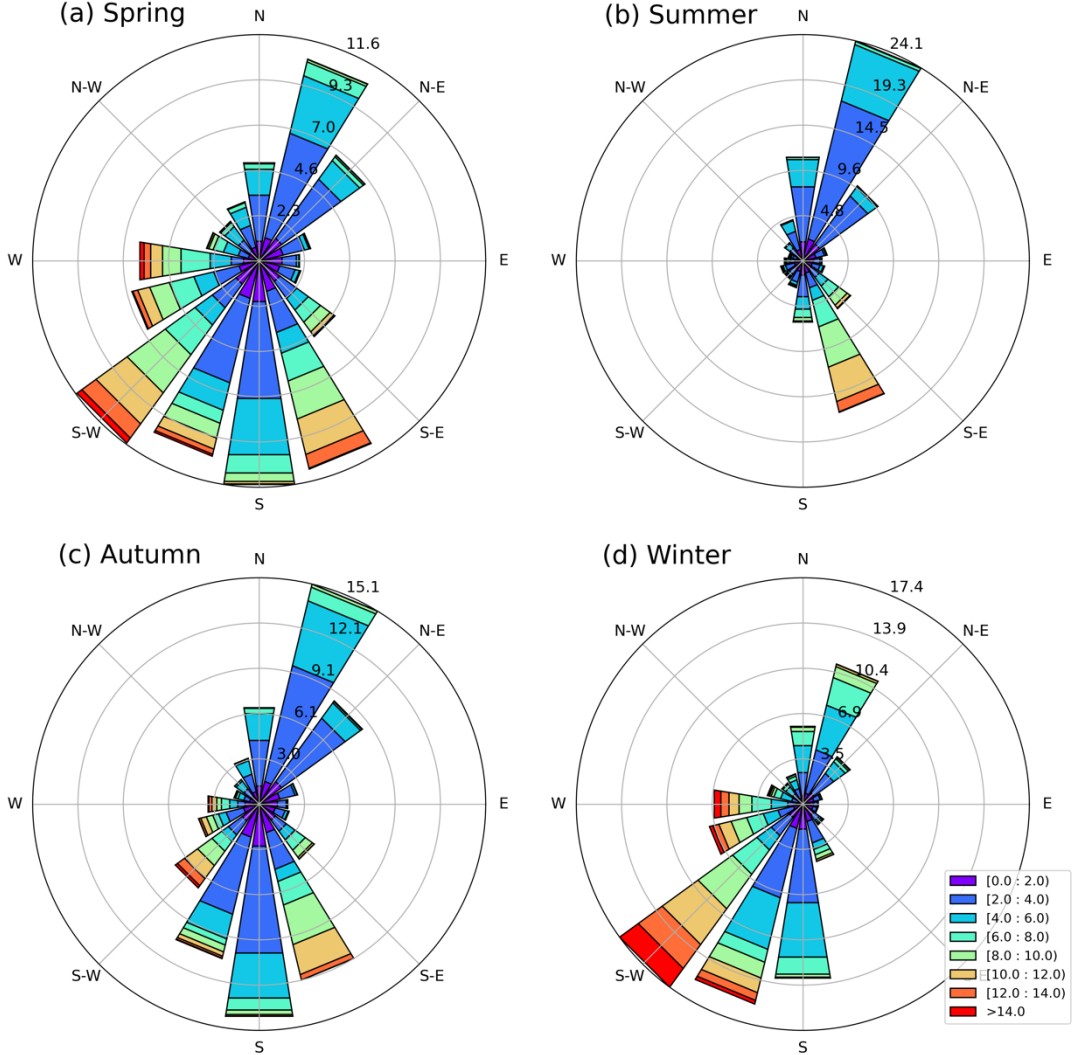

**Figure 3: Seasonal wind roses calculated from the 15-year 10-min wind data at 10 m height at the QOMS station. (a) Spring, (b) Summer, (c) Autumn, and (d) Winter.**

As indicated in Table 1, the continuity of the wind time series is the best at 10 m. Seasonal wind roses calculated from the

15-year 10-min wind time series at 10 m height are shown in Figure 3. Wind roses significantly differ in the four seasons in terms of wind direction and speed. In spring, southerly wind (44.1%, from south-southeast to southwest) and north-northeast wind (10.5%) dominate (Figure 3a). Northerly wind speeds are generally less than 6.0 m/s, while a significant number of southerly wind speeds exceed 8.0 m/s. In summer (Figure 3b), the winds primarily blow from north-northeast (24.1%) and



south-southeast (16.4%) directions, and the south-southeast wind speed is significantly larger than the north-northeast wind

speed. The percentage of wind from other directions is very low. In autumn (Figure 3c), southerly wind (36.8%, from south-southeast to south-southwest) and north-northeast wind (15.1%) dominate. In winter (Figure 3d), southwest wind (17.4%) is predominant, followed by south-southwest wind (15.5%), south wind (13.2%), and north-northeast wind (11.0%). The southwest wind speed is also significantly larger than the wind from other directions. In general, southerly wind and northerly wind are prevailing wind directions at the QOMS station, and the southerly winds are stronger than the northerly

winds. Winter and spring wind speeds are larger than in the summer and autumn seasons. Interestingly, there are almost no west winds in summer. However, the percentage of westerly winds increases, as well as the wind speed gradually increases from autumn to spring. Many studies have reported the phenomenon that the south wind is stronger than the north wind (Cai et al., 2007; Song et al., 2007; Sun et al., 2007; Sun et al., 2017; Sun et al., 2018). Recently, Sun et al. (2018) revealed that the strong south winds in non-monsoon season are dominated by downward momentum transport for westerly winds aloft,

while during the monsoon season, the strong winds are driven by up-valley winds from the Arun Valley east of Mt. Everest migrating into the Rongbuk Valley where the QOMS station is located. Moreover, katabatic winds driven by the along-valley temperature gradient between cold temperatures to the south over glacier surfaces and warm temperatures to the north can accelerate the wind speed.

The monthly average wind speed at 10 m from September 2005 to June 2019 and the annual mean wind speed from 2006 to 2018 are shown in Figure 4. The monthly average wind speed fluctuates mainly between 3.5 and 7.0 m/s and is characterized by pronounced seasonal variations, with high wind speed in winter and low wind speed in summer (Figure 4a). The annual mean wind speed fluctuates slightly around 4.6 m/s. The trend in mean wind speeds from 2006 to 2018 is not detectable, though it seems to exhibit a very slight decreasing tendency (Figure 4b). Over the corresponding time period, the trends in

mean surface wind speed across the entire TP are also not significant (Zhang and Wang, 2020; Zhang et al., 2024).

Predominant wind directions at the QOMS station are slightly different in the four seasons, although the wind blows generally along the river valley. Moreover, wind speeds are significantly different in different directions. Due to the distinct characteristics of wind in different seasons, we analyzed not only the full-scale spectra of wind but also the seasonal spectra

in following sections.




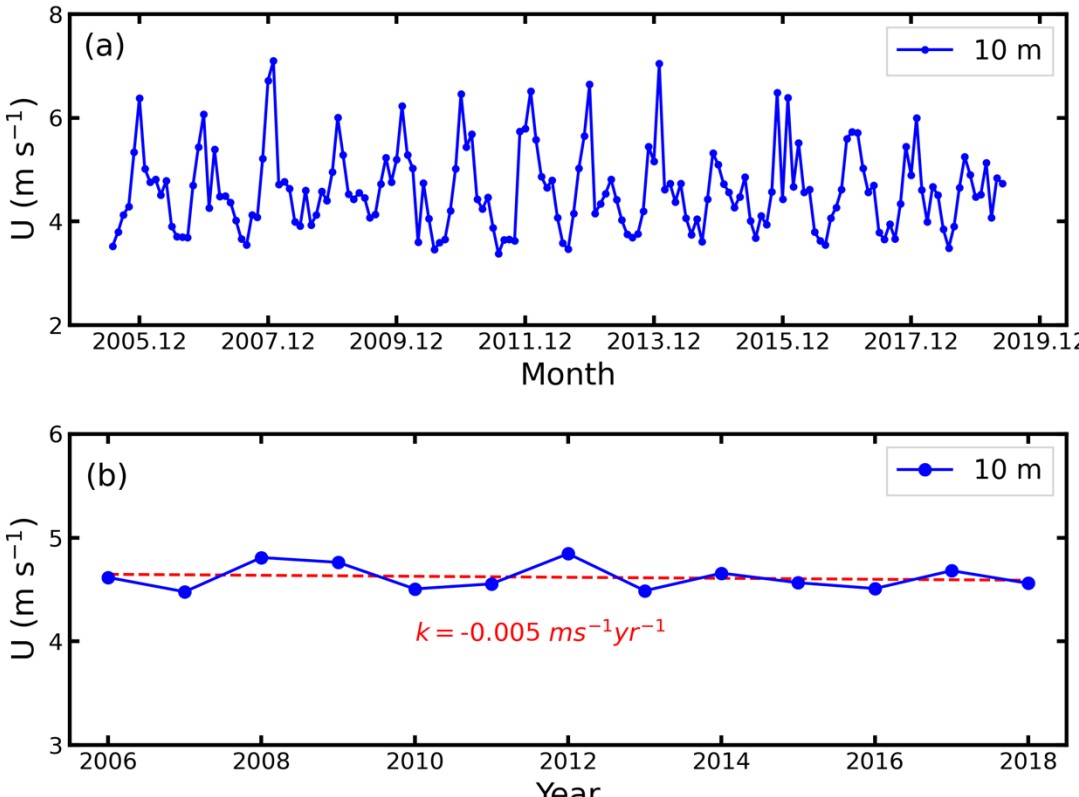

**Figure 4: Time series of monthly averaged horizonal wind speed at 10 m from September 2005 to June 2019 (a) and the annual wind speed from 2006 to 2018 (b). The dashed straight line indicates the linear trend of annual wind speed during 2006~2018, and k is the slope of the straight line.**


## 3.2 Full-scale spectra

As shown in Table 1, the data availability of the 10-min wind data from 1.5 and 20 m was below 90% in multiple years. Therefore, 10-min wind data observed by the PBL tower at 2, 4, 10 m heights were used to calculate the wind speed spectra over the frequency range from about 10 yr$^{-1}$ to 20 min$^{-1}$. As described above, the missing data points were interpolated using the linear interpolation method. Moreover, 10-Hz sonic wind data from 2015 and 2016, which have 100% data coverage on the daily time scale, were used to calculate daily wind speed spectra in the frequency range from about 1 day$^{-1}$ to 5 Hz. Figure 5 shows the full range frequency-weighted spectra $fS(f)$ of wind speed, with $f$ from about 10 yr$^{-1}$ to 5 Hz. Solid curves in Figure 5 are spectra calculated from the 15-year long time series of 10-min horizontal wind speed data, which shows a




more climatologically representative spectrum. The dotted and dashed curves are the composite daily wind spectra calculated
by using the daily 10-Hz sonic wind data observed in 2015 and 2016, respectively.

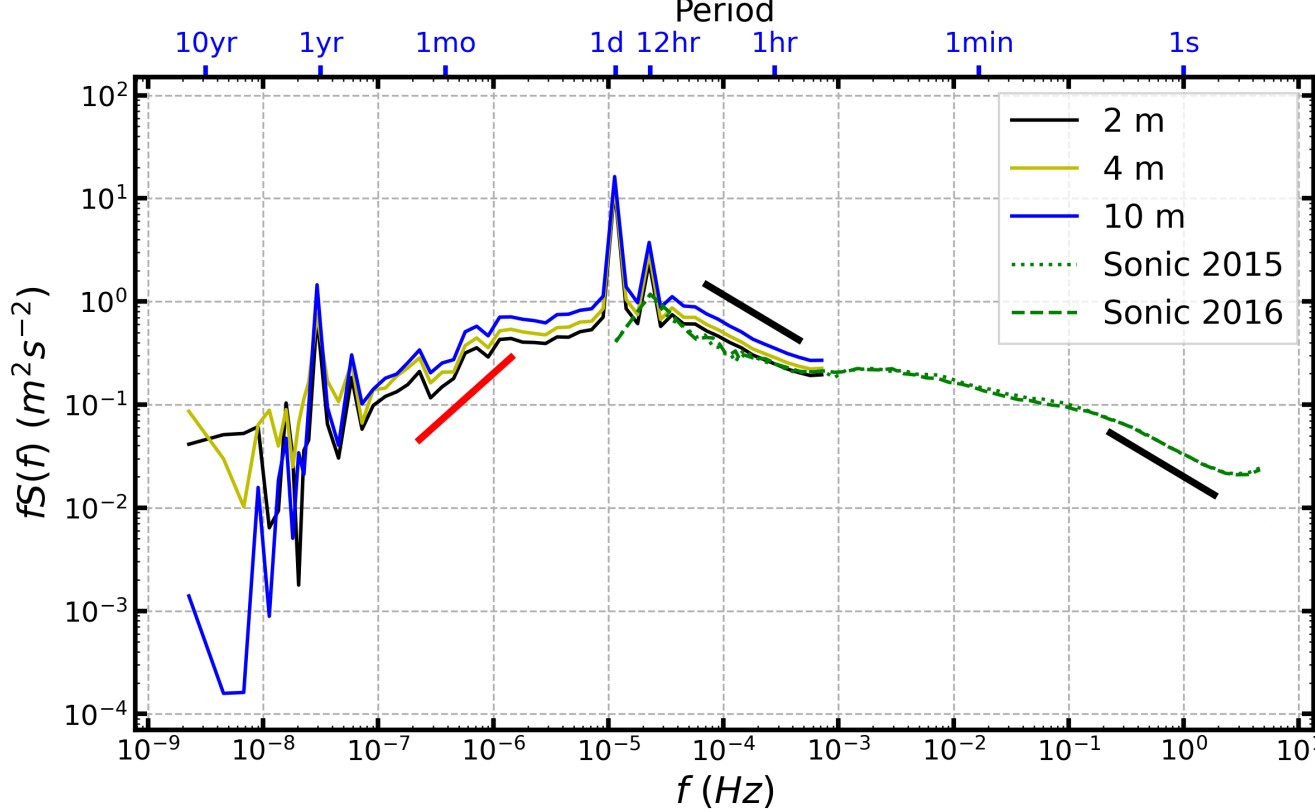

**Figure 5: The frequency-weighted spectra *fS(f)* as a function of frequency f of horizontal wind speed at 1.5, 4, 10, 20 m heights at the QOMS station calculated from the 15-year 10-min wind data (solid curves). The dotted and dashed curves are averaged daily spectra calculated from the 10-Hz sonic wind data from 2015 and 2016, respectively. The thick black lines indicate the reference *f*$^{2/3}$ spectra and the thick red line indicates the reference *f*$^{+1}$.**

In the 15-year full-scale spectra, there are multiple spectra peaks clearly observed. The first narrow peak is at 1 yr$^{-1}$. The yearly peak has been frequently observed in the full-scale spectra at costal, offshore, and terrestrial sites (Kang and Won, 2016; Larsén et al., 2016; Watson, 2019). The second peak is the diurnal peak, which is associated with surface heat flux

modulations. The diurnal peak is often found to be missing or insignificant at an offshore or coastal site, while it is the most significant peak at an inland site. The diurnal peak at the QOMS station is even more significant than previous findings (Horvath et al., 2012; Kang and Won, 2016), the power density of the peak is larger than neighbor power densities by more than one order of magnitude. Interestingly, there is a third peak at the frequency of 12 hr$^{-1}$. The 12 hr$^{-1}$ peak in the wind speed spectrum is expected in mountain area, but it is not always observed. For example, Kang and Won (2016) did not





observe the 12 $hr^{-1}$ peak in their wind speed spectra at a height of 100 m at the BAO site, located 25 km east of the foothills of the Front Range of the Rocky Mountains. However, closer to the ground surface at 10 m, a very weak peak appears to be present (Figure 3b in Kang and Won (2016)). Moreover, many studies have reported a 12 $hr^{-1}$ peak in near-surface horizontal wind velocity in tropical and subtropical oceans, which is attributed to the atmospheric tides (Lindzen and Chapman, 1969; Jacobs, 1980; Ueyama and Deser, 2008; Hsu et al., 2025). The 12 $hr^{-1}$ peak in this study is mainly due to the daytime cycle between morning calm wind and afternoon strong southerly wind, which is the result of interactions between the subtropical westerly jet and local valley winds at the QOMS site (Sun et al., 2018).

The transition between spectra calculated from 10-min wind time series and from 10-Hz sonic wind data is rather smooth, indicating the consistency of the two types of data time series in the overlapping frequency range. The spectra calculated from 10-Hz sonic wind data are almost no difference between 2015 and 2016, which means one year ensemble mean is long enough to have a good climatological representative of the daily spectra covering frequency range from 1 $day^{-1}$ to 5 Hz. Moreover, similar to the 10-min spectra, the 10-Hz spectra also exhibit a peak at the frequency of 12 $hr^{-1}$. Note that the increase in spectral density at the high frequency end of the daily wind spectra (dotted and dashed green curves) is due to the presence of white noise. Similar issues are also present in Figures 6 and 7 in the following sections.



**280   3.3 Seasonal spectra**

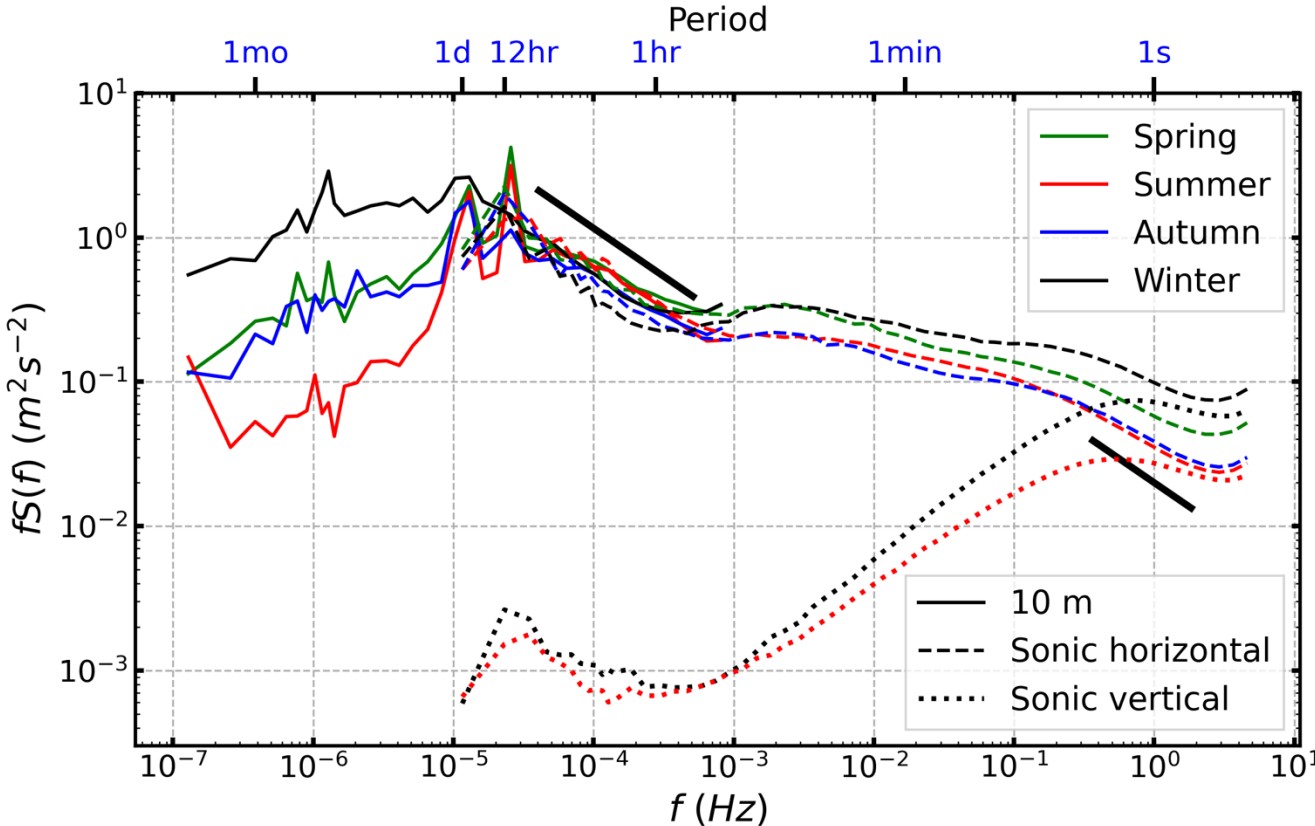

**Figure 6: The frequency-weighted seasonal spectra $f$S($f$) as a function of frequency $f$ of horizontal wind speed at 10 m height at the QOMS station calculated from the 15-year 10-min wind data (solid curves). The dashed curves are averaged daily horizontal wind spectra in four different seasons calculated from the 10-Hz sonic wind data from years 2015 and 2016. The dotted curves are averaged daily vertical wind spectra in summer and winter calculated from the 10-Hz 3D data from years 2015 and 2016. Different colors represent different seasons. The thick black lines indicate the reference $f^{2/3}$ spectra.**

To reveal the seasonal characteristics of the spectra of the 15-year wind speed time series, we divide the year into four seasons according to the following rules: March, April, and May for spring, June, July, and August for summer, September, October, and November for autumn, and December, January, and February for winter. For each season, we first calculated the spectrum of the 3-month wind speed time series. The spectra were then averaged at each frequency over the 15 years to obtain a 15-year composite seasonal spectrum. As data continuity of the 10-min tower data is best at 10 m height, thus we only calculated seasonal wind spectra at this height at the QOMS station. Moreover, 10-Hz sonic wind data from 2015 and 2016 were also used to calculate the seasonal spectra, both horizontal wind and vertical wind spectra, which were then





averaged over the two years. Finally, full-scale seasonal spectra were obtained, covering frequencies from about 3 mo$^{-1}$ to 5 Hz (Figure 6).

As shown in Figure 6, the diurnal peak of the spectra in winter season is less sharp than in other seasons, which is also reported by Kang and Won (2016) using data from the BAO site. Moreover, the 12 hr$^{-1}$ peak is largely absent in winter, while in spring and summer seasons is more significant than the diurnal peak. On the low-frequency side of the diurnal peak ($f \leq 1$ day$^{-1}$), the spectral density in winter is significantly higher than in other seasons. Spring and autumn show similar values, lying between those of winter and summer. In winter, the QOMS station is mainly dominated by strong westerly winds (Figure 3 and Figure 4), and the combination of strong winds and frequent synoptic weather events lead to the high spectral density. By contrast, the low spectral density in summer indicates that synoptic weather events are substantially less frequent in summer than in other seasons.

In the mesoscale frequency range (1 day$^{-1}$ $\leq f \leq$ 1 hr$^{-1}$), the differences in spectral density in the four seasons are not significant as in the low-frequency range. Interestingly, the summer spectral density is comparable to spring and autumn but larger than the winter spectral density in the high-frequency side of the mesoscale frequency range. This suggests more frequent mesoscale activity, which is likely caused by the diurnal variations in wind speed and direction reported by Sun et al. (2018) and by the frequent daytime convective activities in summer in the Mt. Everest region. All frequency-weighted seasonal wind speed spectra ($fS(f)$) fall off as $f^{2/3}$ in the mesoscale frequency range, which seems to be in the Kolmogorov inertial subrange. The winter spectrum falls off even earlier at the frequency of $1 \times 10^{-5}$ Hz. At the frequency around $1 \times 10^{-3}$ Hz, there is a slope transition zone, and the spectral density increases beyond the frequency of $1 \times 10^{-3}$ Hz but with different magnitudes in different seasons. Thus, the transition is the most pronounced in winter, while is obscure and becomes a plateau in summer. In the microscale frequency range ($f \geq 2 \times 10^{-3}$ Hz), spectral densities fall off more smoothly.

In the microscale frequency range ($f \geq 2 \times 10^{-3}$ Hz), spectral densities in winter and spring are substantially larger than in summer and autumn. Winter values are slightly larger than those in spring, while summer and autumn are nearly identical. Generally, near-surface turbulences are generated thermally and mechanically. Högström et al. (2002) reported that turbulence at frequencies higher than $10^{-3}$ Hz is generated mainly by wind shear in the surface layer between the ground surface and the measurement height. As indicated in Figure 4a, wind speed in winter is significantly larger than summer. Moreover, stronger winds would produce higher frequencies as the relationship between scale in m and frequency in Hz is often a function of the windspeed. Thus, the strong wind speed and wind shear explain the high spectral density in winter. Kang and Won (2016) investigated seasonal spectra using wind data observed at the BAO facility in Erie, Colorado. They found that in the microscale frequency range ($f \geq 2 \times 10^{-3}$ Hz), spectra densities in spring and summer were slightly larger than





those in winter and autumn. The differences in seasonal spectra between this study and the results of Kang and Won (2016) are likely due to variations in climatic conditions, land surface characteristics, and topography. Moreover, daily vertical wind spectra calculated from the 10-Hz sonic wind data for summer and winter are also plotted in Figure 6. Unlike the horizontal wind spectra, which decrease in the frequency range from approximately $2\times10^{-3}$ Hz to $3\times10^{-1}$ Hz, the vertical wind spectra increase monotonically. At higher frequencies ($f \geq 1\times10^{0}$ Hz), the shape of the vertical wind spectra is similar to the horizontal wind spectra, with spectral densities that are very close as well.

**3.4 Spectra gap**

In the transition range between mesoscale and microscale, at frequencies around $1.0\times10^{-3}$ Hz, a spectral gap is frequently observed (Van der Hoven, 1957; Fiedler and Panofsky, 1970; Högström et al., 2002; Larsén et al., 2016). Due to the limited understanding of the interactions between 2D and 3D turbulences in this transition range, the existence of the gap remain a subject of debat. In this study, an obvious spectral gap between $1.0\times10^{-4}$ and $1.0\times10^{-3}$ Hz exists in both the 15-year full-scale spectra (Figure 5) and seasonal spectra (Figure 6).







**Figure 7: The frequency-weighted summer and winter daily spectra $fS(f)$ as a function of frequency f of horizontal wind speed calculated from the 10-Hz sonic wind data collected in 2015 and 2016. The thick black and blue dashed lines indicate Equation (1) with different a1 values. The thick blue and red lines indicate the reference $f^{2/3}$ and $f^2$ spectra, respectively.**

As the wind speed, wind direction, and spectral characteristics significantly differ between summer and winter seasons, daily spectra in these two seasons were selected to investigate the spectral gap in terms of its generality and behavior. The 10-Hz sonic wind data collected in 2015 and 2016 were used to the calculate the daily spectra, and only days without missing data points were selected. Figure 7 shows the frequency-weighted summer and winter daily spectra of horizontal wind speed. There is pronounced spectral density minima at $4.5\times10^{-4}$ Hz in winter spectrum, indicating the spectral gap which is also the transition from mesoscale 2D flow to 3D boundary-layer turbulence. The spectral gap almost vanished in the summer spectrum. The spectral density in summer is higher than in winter for frequencies lower than the spectral gap, while above the gap, the spectral density in winter increases significantly and surpasses that of summer. This results in a more noticeable



gap in winter than in summer. The frequency range higher than $1.0\times10^{-3}$ Hz represents the classical 3D boundary-layer turbulence region and the frequency range $5.0\times10^{-3} < f < 2.0\times10^{-1}$ Hz is known as the shear production range (Tchen et al., 1985; Högström et al., 2002; Larsén et al., 2016). Consequently, the turbulent activity, especially the shear production, is more pronounced during winter than in summer at the QOMS site.

The spectral structure at frequencies lower than the spectral gap has been less studied and understood (Högström et al., 2002; Larsén et al., 2013; Larsén et al., 2016). By analogy to Lindborg (1999)'s equation, Larsén et al. (2016) proposed a model to express the spectral structure in the mesoscale range that the weighted spectrum $fS(f)$ decreases with frequency with a $f^2$ slope and followed by a $f^{2/3}$ slope as

$$S(f) = a_1 f^{-2/3} + a_2 f^{-2}, \tag{1}$$

with $a_1 = 3\times10^{-4}$ m$^2$s$^{-8/3}$ and $a_2 = 3\times10^{-11}$ m$^2$s$^{-4}$. Horizontal wind speed spectra observed at coastal onshore and offshore sites have been proved to match the spectral model perfectly (Larsén et al., 2013; Larsén et al., 2016). Regarding the applicability of Equation (1) to spectra in mountainous regions, they suggested that further examination is needed. Equation (1) is plotted as the dashed blue line in Figure 7 but with $a_1 = 9\times10^{-4}$ m$^2$s$^{-8/3}$ and $a_2 = 3\times10^{-11}$ m$^2$s$^{-4}$, which describes the winter spectra in the mesoscale frequency range very well. For the summer spectra, $a_1 = 15\times10^{-4}$ m$^2$s$^{-8/3}$ needs to be used and $a_2$ remains the same. Moreover, a reference line with a slope of -2/3 on a log-log scale, representing the theoretical spectral trend, is also plotted in Figure 7. As shown, the two lines plotted based Equation (1) are parallel to the reference line. This means the second term on the right-hand side of Equation (1) is negligible and the spectra converge to $f^{2/3}$ (figures omitted). Similar results were reported by Kang and Won (2016) as well. Therefore, the frequency-weighted power spectrum in the mesoscale frequency range (1 day$^{-1} \leq f \leq 1$ hr$^{-1}$) essentially follows a -2/3 slope on a log-log scale.

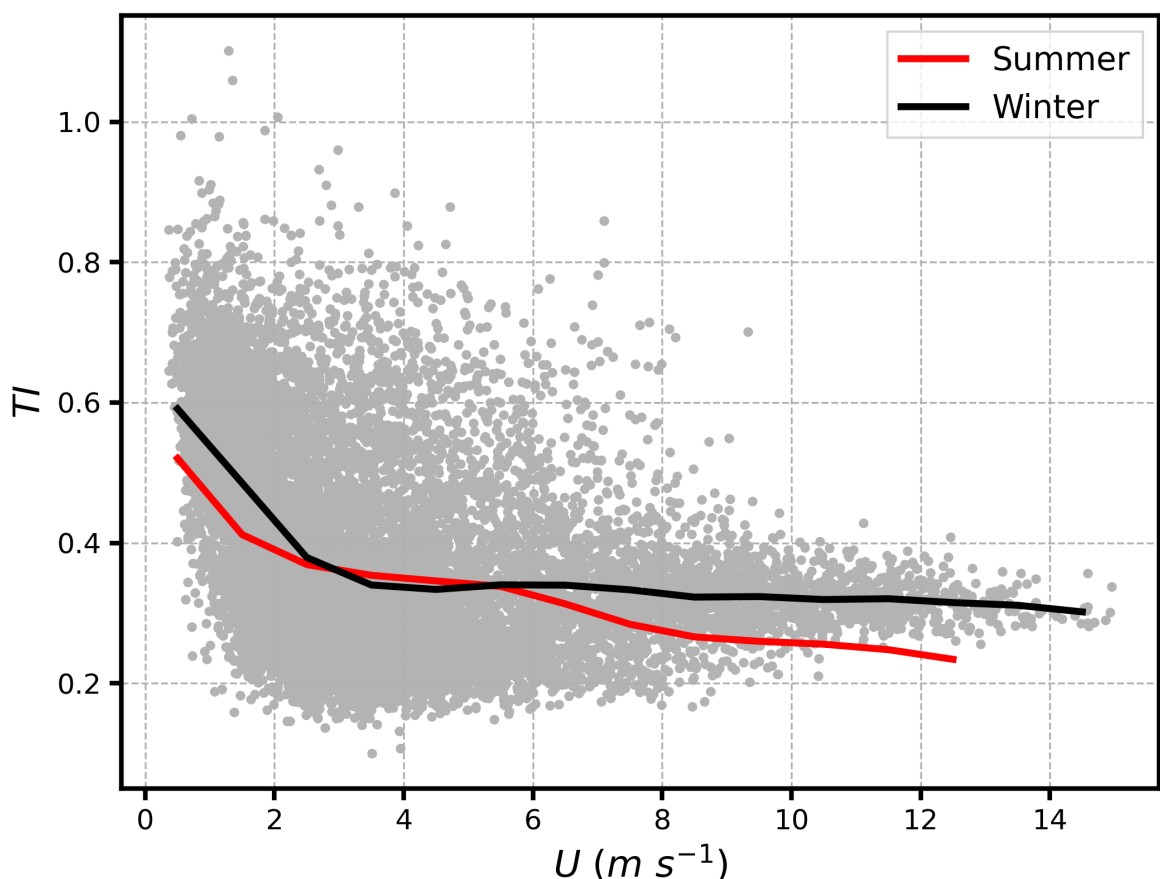

**Figure 8: Turbulence intensity (TI) as a function of horizontal wind speed. The gray dots represent TI values in winter 2016, calculated over 30-minute segments from the daily time series of 10-Hz sonic wind data. The red and black lines are mean value of**
**TI in summer and winter 2016, respectively.**

To better understand the seasonal differences in the turbulence gap between winter and summer, we calculated the turbulence intensity (TI). The turbulence intensity is defined as

$$TI = \sqrt{\sigma_u^2 + \sigma_v^2}/U, \tag{2}$$

$\sigma_u$ and $\sigma_v$ are standard deviation of wind speed $u$ and $v$, respectively; $U$ is the mean values of horizontal wind speed. As the calculation of TI is sensitive to the averaging time period, a 30-minute segment was used in this study. Figure 8 shows TI varying with horizontal wind speed for summer and winter in 2016. It is evident that the range of wind speed is broader in



winter than in summer, indicating that wind gusts are stronger in winter and consistent with the higher mean winter wind speed reported in Section 3.1. Overall, TI values are also larger in winter than that in summer, which again indicates that 
turbulence is stronger in winter than in summer. This is most likely driven by enhanced near-surface turbulent activities associated with the downward transport of large-scale westerly momentum during winter (Lai et al., 2021). Thus, high spectral energy density associated with large TI in the microscale frequency range in wintertime, indicating a dynamically energetic flow regime with strong turbulence production and mixing efficiency. In combination with the lower turbulence spectral density in winter compared to summer in the mesoscale frequency range, these factors contribute to a more 
pronounced spectra gap in winter.

## 4 Summary and conclusions

A 15-year time series of horizontal wind speeds observed at a mountainous site in a valley 30 km north of the peak of Mt. Everest was used to investigate the full-scale spectra of horizontal wind speed. The characteristics of wind speed and wind speed spectrum were analyzed. The findings of this study advance the understanding of the wind speed spectrum in 
mountainous regions. These results can help refine high-resolution atmospheric models and enhance understanding of local circulation dynamics.

About 71.5 % of the winds blow along the direction of the river valleys at the QOMS station. Wind speed shows significant seasonal variation, with stronger speeds in winter and weaker speeds in summer. From 2006 to 2018, the annual average 
wind speed shows almost no detectable trend, although exhibiting a very slight decreasing tendency. This trend is consistent with results reported for other regions, where no significant increase or decrease in near-surface wind speed over the TP has been observed since 2002.

In the 15-year full-scale spectra, we identified three peaks at frequencies around 1 yr$^{-1}$, 1 day$^{-1}$ and 12 hr$^{-1}$, respectively. The 
12 hr$^{-1}$ peak in the wind speed spectra is significant in spring and summer, while it disappears in winter. The spectral peak at 12 hr$^{-1}$ is most likely related to the daytime local circulations in the valley where the QOMS station is located. The seasonal spectra show distinct characteristics in different seasons. On the low-frequency side of the diurnal peak ($f \leq 1$ day$^{-1}$), the spectral density is the highest in winter, which is associated to the strong westerly wind and frequent synoptic weather events during the winter season. On the high-frequency side of the spectral gap, the spectra density in winter and spring is much 
higher than in summer and autumn, suggesting that at the QOMS station, the turbulence intensity, especially that generated by shear, is more significant in winter than in summer. This is due to the strong winds during the winter and spring seasons.

A spectral gap is observed between the frequency of $1.0 \times 10^{-4}$ and $1.0 \times 10^{-3}$ Hz. It is the most obvious in the winter season while almost vanished in summer. The spectral density in the mesoscale range is lower in winter than in summer, while the

spectral density in the microscale range is higher in winter than in summer, leading to the most pronounced spectral gap during winter. Moreover, turbulence intensity is greater in winter than in summer due to the downward transport of westerly momentum, further contributing to this pronounced spectral gap.

**Acknowledgments**

This project has received funding from the National Key R&D Program of China (grant no. 2022YFB4202104) and the

National Natural Science Foundation of China (grant nos. 42230610, 41975013). We gratefully acknowledge the colleagues at the QOMS site for their diligent maintenance of the instruments.

**Code/Data availability**

All the data used in this study have been published in Ma et al. (2020) and are publicly accessible at the National Tibetan Plateau Data Center ([https://doi.org/10.11888/Meteoro.tpdc.270910](https://doi.org/10.11888/Meteoro.tpdc.270910), last access on August 20, 2025). The codes for data

processing are available upon request.

**Author contribution**

Conceptualization: CH, YM, WM.
Methodology: CH, FS, YZ.
Visualization: CH.
Funding acquisition: CH, YM, FS.
Writing (original draft): CH.
Writing (review and editing): YM, WM, FS, YZ, HX, WH, CD, ZX.

**Competing interests**

The authors declare that they do not have any competing interests.





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
