# Peer review of "Full-scale spectra of 15-year time series of near-surface horizontal wind speed on the north slope of Mt. Everest"

_EGUsphere, 2025_

## Referee Comment (RC1)

Review of: Full scale spectra of 15-year time series of near-surface horizontal wind speed on the north slope of Mount Everest, by Han et al.

Title: The station is in a valley that significantly affects the winds, rather than on the north slope of Mount Everest" as seen from Figure 1. Perhaps in a valley on the north SIDE of Mt Everest."

**General Comments**

While some interesting results are suggested in places, this paper contains some significant errors and internal contradictions, discussed in more detail below.

- (1) Figures 6 and 7 show vastly different sonic-anemometer-based spectra; and extension of sonic data to frequencies much larger than it was designed for.
- (2) The authors use a relationship (equation (1)) that absurdly assumes a universal spectrum for high and low frequencies, and then uses it to provide the dashed slope lines in Fig. 7).
- (3) After explaining the more energetic high-frequency spectrum in winter in terms of larger wind speed, the authors produce a graph (Fig. 8) that shows the highest turbulence energy at the lowest wind speeds.

Also, while the role of Everest seems to be focused on, the real physics has to do with the terrain surrounding the site. This could be addressed by either expanding the area in Fig. 1 so that the fetch from all directions is better represented, or better yet, by adding a figure that focuses on the terrain in all directions nearer the site. Some discussion already appears; it would be enriched by such a figure.

Furthermore, processing of sonic-anemometer data can be quite involved. There is nothing about correction for orientation, etc. that usually is addressed in a paper dealing with sonic data. Such care could improve the slope at higher frequencies except of course for the aliasing at the highest frequencies. I am unsure about whether sonic data have been used at the lowest frequencies.

Thus, this paper needs a major rework, and probably redoing some data processing and analysis, before it can be considered for publication.

**Specific Comments**

**Abstract:**

Again, north SIDE of Mt. Everest (L 19). "slope" implies that the station is not in a valley.

L78. This is incorrect. Sampling rolls in the cross-roll direction leads to one wavelength but the gap between synoptic-scale energy and roll-scale energy will still exist. (though it is unlikely that an aircraft sampling rolls would have a track long enough to sample synoptic variability).

Furthermore, if the boundary layer eddies are being sampled from a tower, the rolls will show up with a period of the order of a half-hour or more, with large eddies with a period of the order of five minutes (at the strong wind speeds associated with rolls). So there will be a gap between the signal representing the large eddies and the signal representing the rolls, and a gap between the signal of the rolls and synoptic variation.

Also, if the winds are weak, large eddies could have long period as well – even of the order of a half hour, assuming the large eddies aren't evolving. I.e., things get complicated. But there could still be a gap between the large eddy signal and the synoptic signal.

- L85-87. This sentence provides no information how could two sites show "universal characteristics consistent with findings in the literature." Please correct or delete.
- L94. Re Li et al. (2021). Horizontal wind component spectra? Wind speed spectra? The results could be different for spectra of wind speed and spectra for one of the wind components.
- L100. GATE needs to be spelled out. It is the Global Atmospheric Research Programme (GARP) Atlantic Tropical Experiment.
- L105. Delete starting with "therefore".
- L112-L115. Discussion of local circulations. This discussion is not clear, but it is important since it appears to apply to where the measurements are taken. Why is it so different from idealized mountain-valley circulations? I have a hunch that it has to do with the side of Mt. Everest the valley is on. This should be discussed in terms of which slopes are illuminated by sunlight, and which slopes are not. I.e., the surroundings in all directions should be considered.

This discussion could be greatly improved through use of a second map showing the local terrain surrounding the site. That is, it shouldn't cover as large an area as Fig. 1.

- L119. Regarding the next sentence about being "debated" perhaps it is more precise to say that the origins of circulations in the Himalaya are still the subject of active research.
- L123. Since the station is in a valley, which profoundly affects the winds, one should be more precise, e.g.., what is being documented is 'the winds in a valley on the north side of Mt. Everest, 30 km to the north of the peak.' Further, this provides insight into the local circulation there. ('on the north slope of Everest' implies being ON the slope, which would imply a different circulation than one in a valley on the north side.). Which is also affected by other nearby terrain.
- L137. Interaction between the atmosphere, terrain, and surface cover?
- L138. Suggest, "The QOMS station is situated at the bottom of the Rongbuk Valley at an elevation of 4276 m above sea level, approximately 30 km north of the summit of Mt. Everest (Figure 1). At the station, the Rongbuk Valley runs in a NNE-SSW direction (approximately 10° to 190°), with a width of about 1.5 kilometers.

(NOTE: 10 to 190 degrees is closer to north-south than to NNE-SSW, but the map suggests something closer to NNE-SSW, which would be more like 22.5 degrees to 202.5 degrees). PLEASE CHECK.

- L145. Wind direction and speed?
- L151. Should read 'wind speed and direction' (velocity is a vector). Though, didn't you simply analyze wind speed? You need to be specific about this. People either analyze wind components

(u,v,w), horizontal wind components (u,v) or wind speed; and the results obtained for wind speed vs u and v are different. Okay, it's included in the next paragraph.

Table 1. 'Total' should be 'Total Days'

Section 2.3 okay, although it is important to document how the sonic-anemometer data were processed (corrections for orientation, etc.) and then analyzed.

Figure 2. I would have expected the winds going downslope from Everest would provide the dominant direction, which would be from the SSW. But I am not sure what the terrain looks like to the north of the site. So – when you write 'northeasterly to northerly' do you mean 'toward the northeast to north'? (This is one of the frustrating ambiguities of the English language!).

I just looked up "wind rose" on the Internet, which indicates that wind roses indicate the direction from which the wind is blowing, so I am guessing I should interpret the wind is mostly being from the NNE. If so, I would recommend more explicitly writing "north to northeast winds" or even more specifically, "winds from the north to northeast."

Assuming that the dominant wind directions are from the north to northeast, it appears that nearby terrain or terrain to the north has a strong influence as well.

The following sentences, which refer to north winds, suggest that most the winds are from the north-northeast, but I think it would be better to avoid use of 'easterly', etc.

On a related matter, since you are presenting a complete wind rose, it would be useful to have a topographic map with the station at the center. Could be a simplified version. (see other comments about this)

L213-L215. Suggest, for more precision and less ambiguity.

In general, prevailing winds are along the valley, with directions from the SSE to SW in the spring, autumn and winter; and from the NNE in summer; with higher speeds from the southern directions.

L218-220. This sentence makes no sense. Why would downward transport of zonal wind (from the west) increase the winds from the south? Please check and correct.

L220-221. For clarification, suggest (If this is correct)

The strong SSW to SW winds in Fig. 3 are driven by .. (rest of sentence okay)

L245. Insert 'for the days"

L250 The existence of aliasing at the highest frequencies should be announced immediately, in Section 2.3, BEFORE the first figure showing spectra.

I.e., Lines 278-280 be deleted, replaced with a statement in Section 2.3 that the increase in spectral energy at the highest frequency is due to aliasing, which results from sampling a signal at too low a rate. This applies to all the spectra shown. It has nothing to do with white noise.

L258. "costal" should be 'coastal'.

L260. How about simply contrasting "over land" vs "over the ocean." And, it would be more precise to say the average diurnal peak is smaller over the ocean, rather than 'missing.' You could check whether the Uemaya paper shows a weak diurnal cycle, so that could be cited. A diurnal peak over land is common knowledge, so no citation would be necessary,

Regarding the 12-h peak in a mountain area, it would be useful to cite the presence of thermally-driven upslope and downslope winds, citing the book by Whiteman. (I think 'upslope and downslope' winds is sufficiently broad to include upvalley and downvalley winds as well). You could use the adjective "common" or "frequently seen" to indicate that they would not always be observed, since well-defined thermally-driven circulations are most probable in fair-weather, weak-synoptic wind, situations.

L264. Since the Kang and Wan papers DOES observe a weak 12-hr peak at the Boulder Atmospheric Observatory, contradicting rather than supporting your statement. I would omit citation of this study here. Also, the measurements are not in a mountain area, but near a mountain area.

L269-271. There is missing information here. You cannot explain a 12-hour cycle showing up that well with one maximum and one minimum in wind speed in 12 hours.

**L294. Horizontal wind speed?**

Comparing Figs 6 and 7 – In Fig. 6, the frequency-weighted sonic spectral energy for horizontal wind is three orders of magnitude lower than that found from the measurements from the tower at 10 m. Yet in Fig. 7, you the sonic energy of the same order of magnitude as that found at the tower at the low-frequency end!

Similarly, the slope of the sonic-derived spectral energy is positive in Fig. 6 over several decades where it is negative in Fig. 7.

Something is wrong here! Data processing? Even believing sonic data at such low frequencies?

L303-4. Figures 3 and 4 show wind SPEED not the west-to-east wind component, so this statement needs correcting (or the figure needs correcting). Also, you should make sure the cited papers in line 229-230 are for the corresponding measurement.

L310. The mesoscale motions being related to the complex terrain (e.g., mountain-valley flows) and deep convection?

L313. But the basic physics of the Kolmogorov frequency range assumes the smaller eddies simply get their energy from the larger eddies. In this frequency range, the motions are driven by buoyancy associated with nonuniform heating (for complex-terrain flows) and release of latent heat (deep convection). So, it's coincidence. The curious thing is the lack of a -2/3 slope at the higher frequencies, where it's expected (even though there is some energy introduced at higher frequencies).

L323. Simply because eddies pass by the sensor faster?

L331-332. I question whether the spectra are reliable at these frequencies, due to a combination of aliasing and a significant departure from the expected -2/3 slope.

L335 – L339. The existence of a spectral gap is not necessarily a matter of debate; rather the existence of such a gap varies with the situation (and sometimes the method of sampling).

Figure 7. Lots of problems here! (1) Why is the low-frequency sonic-derived spectral energy three orders of magnitude here higher than in Fig. 6? (2) Moreover, even the slope of the spectral energy as a function of frequency is of opposite sign between 10-3 and 10-1 Hzm=. (3). Eq (1) is arbitrary and contributes nothing, so fitting the low frequencies to match the data has no meaning. (4) Thus, you need only one thick line, to indicate slope – the solid blue line. The red line isn't needed, unless you want to state that the low frequencies don't look like 2-D turbulence. ((1) and (2) noted earlier)

L349. The spectral gap DID vanish.

L350, surpasses that of summer, as noted earlier. Can remove the text after "summer" as this was already discussed in detail.

(I am ignoring discussion of the dubious eq (1))

L376-L380, figure 8. Something is very weird about this plot. It suggests that turbulence intensity is higher with weaker wind, quite the opposite of what was discussed earlier. I suggest that this section either be corrected or deleted altogether.

L405-406. For the 12-h spectral peak, should check nighttime winds, too, as noted before.

414. There is no spectral gap in summer.

---

## Author Comment (AC2)

Response to Referee's comments:

We would like to thank the Editor and the Referee for the time and efforts handling the reviewing our manuscript. The constructive comments and suggestions were very helpful to improve the manuscript.

The Referee's original comments are formatted in black, while our point-by-point responses are formatted in **blue** font. All the corresponding revisions in the revised manuscript are indicated using the "Track Changes" function in the Word document.

**Reviewer #1**

**Review of: Full scale spectra of 15-year time series of near-surface horizontal wind speed on the north slope of Mount Everest, by Han et al.**

Title: The station is in a valley that significantly affects the winds, rather than on the north slope of Mount Everest" as seen from Figure 1. Perhaps 'in a valley on the north SIDE of Mt Everest."

Thank you for your suggestions. The title has been changed to "*Full scale spectra of 15-year time series of near-surface horizontal wind speed **in a valley on the north side** of Mount Everest*".

**General Comments**

While some interesting results are suggested in places, this paper contains some significant errors and internal contradictions, discussed in more detail below.

(1) Figures 6 and 7 show vastly different sonic-anemometer-based spectra; and extension of sonic data to frequencies much larger than it was designed for.

Thank you for your comments and suggestion. We apologize if our description or figures were not sufficiently clear and may have led to some confusion. In fact, **in Figure 6, the four solid curves** show frequency-weighted seasonal spectra of horizontal wind speed (10-min wind time series) measured at 10 m in four different seasons. **The four dashed curves in Figure 6** show daily averaged spectra of horizontal wind speed calculated from the 10-Hz sonic-anemometer data in four different seasons. **The two dotted curves in Figure 6** show daily averaged spectra of vertical wind speed in summer (red dotted) and winter (black dotted) calculated from the 10-Hz sonic-anemometer data. **The red (summer) and black (winter) solid**

**curves in Figure 7** are also the frequency-weighted daily spectra of horizontal wind speed calculated from the 10-Hz sonic-anemometer data.

As shown in Figure 6 (the dashed curves) and Figure 7 (the two solid curves), the daily averaged spectra of horizontal wind speed are consistent in terms of values and structure varying with frequency. Moreover, as shown in Figure 6, the spectra calculated from 10-min wind speed and from 10-Hz sonic-anemometer data agree well with each other, indicating good consistency of the two types of observations in the overlapping frequency range.

Regarding the slope of spectral energy, slopes of horizontal wind speed spectra in Figure 6 and Figure 7 are all negative. The slopes of vertical wind speed spectra in Figure 7 are positive (the red and black dotted curves).

Please also see our responses to your specific comments related to this concern.

(2) The authors use a relationship (equation (1)) that absurdly assumes a universal spectrum for high and low frequencies, and then uses it to provide the dashed slope lines in Fig. 7).

Thank you for your comments and suggestions. We have deleted all content related to Equation (1) in the revised manuscript.

(3) After explaining the more energetic high-frequency spectrum in winter in terms of larger wind speed, the authors produce a graph (Fig. 8) that shows the highest turbulence energy at the lowest wind speeds.

Thank you for your comments and suggestions. We apologize if our description was not sufficiently clear and may have led to some confusion. In fact, in the atmospheric boundary layer, turbulence intensity ($TI = \sqrt{\sigma_u^2 + \sigma_v^2}/U$) generally decreases with increasing mean wind speed. This is because as wind speed increase, although turbulent fluctuations ($\sigma_u$, $\sigma_u$) may still intensify, their relative contribution decreases, leading to a decreasing trend of TI with wind speed. This phenomenon has been discussed in Panofsky and Dutton (1984) and Stull (1988), and has been observed in Ernst and Seume (2012) and Kang and Won (2016).

In Figure 8, the black line shows the mean value of TI in winter 2016, while the red line shows the TI in summer. Here we would like to show that the TI in winter is higher than that in summer, as the wind speed is stronger in winter than in summer (Figure 3 and Figure 4). Moreover, the spectral density in winter is also higher than in summer in the microscale frequency range (figure 6). Thus, wind speed, spectral density, and turbulence intensity discussed in the study are consistent with each other.

References:

1. Ernst, B. and Seume, J. R.: Investigation of Site-Specific Wind Field Parameters and Their Effect on Loads of Offshore Wind Turbines, Energies, 5, 3835-3855, 2012.
2. Kang, S.-L. and Won, H.: Spectral structure of 5 year time series of horizontal wind speed at the Boulder Atmospheric Observatory, Journal of Geophysical Research: Atmospheres, 121, 11,911-946,967, 2016.
3. Panofsky, H. A. and Dutton, J. A.: Atmospheric turbulence: Models and methods for engineering applications, John Wiley & Sons, New York, 1984.
4. Stull, R. B.: An introduction to boundary layer meteorology, Kluwer Academic Publishers, Dordrecht, 1988.

Also, while the role of Everest seems to be focused on, the real physics has to do with the terrain surrounding the site. This could be addressed by either expanding the area in Fig. 1 so that the fetch from all directions is better represented, or better yet, by adding a figure that focuses on the terrain in all directions nearer the site. Some discussion already appears; it would be enriched by such a figure.

Thank you for your comments and suggestions. We have revised Figure 1 and added a topographic map covering an area of $10\times10$ km$^2$ with the QOMS station at the center. Please see Figure 1b in the revised manuscript.

Furthermore, processing of sonic-anemometer data can be quite involved. There is nothing about correction for orientation, etc. that usually is addressed in a paper dealing with sonic data. Such care could improve the slope at higher frequencies except of course for the aliasing at the highest frequencies. I am unsure about whether sonic data have been used at the lowest frequencies.

Thank you for your comments and suggestions. Following the reviewer's suggestion, we applied a double rotation correction to all sonic-anemometer data before further analysis in the revised manuscript. In the original version, we did not apply the double rotation correction to the sonic-anemometer data as the correction does not affect the calculation of the horizontal wind speed spectrum. However, it does affect the calculation of the wind components (u, v, w) spectra. Please see the plots (Figure 1) in our response to your specific comments, which show the comparison before and after the double rotation correction.

The data processing was also introduced in the revised manuscript. Please see lines 179-184 at the end of the first paragraph of Section 2.3 in the revised

manuscript: "……*Before the calculation of wind speed spectra, the 10-Hz 3D sonic wind data were processed using a double coordinate rotation. In this procedure, the wind components were first rotated around the vertical axis to align the streamwise direction with the mean wind and then rotated around the lateral axis to remove the mean vertical velocity. The main purpose of this treatment is to correct for sensor tilt and local terrain effects, ensuring that the horizontal and vertical wind components are physically consistent and suitable for turbulence and spectral analyses……*"

Thus, this paper needs a major rework, and probably redoing some data processing and analysis, before it can be considered for publication.

Following your comments and suggestions, we have carefully revised the manuscript and responded point by point to all three general comments as well as all specific comments. Most importantly, we applied the double coordinate rotation correction to all sonic-anemometer data and reanalyzed the results. Thank you again for your helpful comments, and we hope that our revisions adequately address your concerns.

**Specific Comments**

Abstract:

Again, north SIDE of Mt. Everest (L 19). "slope" implies that the station is not in a valley.

"north slope of Mt. Everest" has all been changed to "north side of Mt. Everest" in the entire manuscript.

L78. This is incorrect. Sampling rolls in the cross-roll direction leads to one wavelength but the gap between synoptic-scale energy and roll-scale energy will still exist. (though it is unlikely that an aircraft sampling rolls would have a track long enough to sample synoptic variability).

Furthermore, if the boundary layer eddies are being sampled from a tower, the rolls will show up with a period of the order of a half-hour or more, with large eddies with a period of the order of five minutes (at the strong wind speeds associated with rolls). So there will be a gap between the signal representing the large eddies and the signal representing the rolls, and a gap between the signal of the rolls and synoptic variation.

Also, if the winds are weak, large eddies could have long period as well – even of the order of a half hour, assuming the large eddies aren't evolving. I.e., things get complicated. But there could still be a gap between the large eddy signal and the

synoptic signal.

*Thank you very much for your detailed explanation. We agree with you. The sentence has been deleted in the revised manuscript.*

L85-87. This sentence provides no information – how could two sites show "universal characteristics consistent with findings in the literature." Please correct or delete.

*The sentence has been deleted.*

L94. Re Li et al. (2021). Horizontal wind component spectra? Wind speed spectra? The results could be different for spectra of wind speed and spectra for one of the wind components.

*It is the horizon wind speed. The sentence has been changed to "……full-scale spectra of multilayer horizontal wind speed……" in the revised manuscript.*

L100. GATE needs to be spelled out. It is the Global Atmospheric Research Programme (GARP) Atlantic Tropical Experiment.

*Thanks for your comments. Full name of GATE has been given in the revised manuscript.*

L105. Delete starting with "therefore".

*The sentence has been removed in the revised manuscript.*

L112-L115. Discussion of local circulations. This discussion is not clear, but it is important since it appears to apply to where the measurements are taken. Why is it so different from idealized mountain-valley circulations? I have a hunch that it has to do with the side of Mt. Everest the valley is on. This should be discussed in terms of which slopes are illuminated by sunlight, and which slopes are not. I.e., the surroundings in all directions should be considered.

*Thank you for your comments and suggestion. The observation site in that study (Sun et al., 2007; Sun et al., 2018) is the same as the one used in our research. We have added relevant information about the site location and the surrounding terrain in the revised manuscript. At this site, sunlight is obstructed by the terrain in the east–west direction, while it is largely unobstructed in the north–south direction. Please check the following sentences in the revised manuscript: "……However, observations at the National Observation and Research Station for Qomolongma Special Atmospheric Processes and Environmental Changes (QOMS), the same observation site as in this study, frequently show strong up-valley winds from noon until sunset and weaker up-valley winds at night, while weak down-valley winds occur from*

*sunrise to noon time (Sun et al., 2007; Sun et al., 2018). The QOMS station is located at the bottom of the Rongbuk Valley, which runs in an NNE-SSW direction and has a width of about 1.5 kilometers at the station (as indicated in Fig. 1). The mountains to the east and west block sunlight to the observation site when the solar elevation angle is low, whereas the mountains to the north and south cause no obstruction……"*

This discussion could be greatly improved through use of a second map showing the local terrain surrounding the site. That is, it shouldn't cover as large an area as Fig. 1.

Thanks for your suggestion. An additional map showing the terrain surrounding the QOMS site was added to Fig. 1. Please check Fig. 1b in the revised manuscript.

L119. Regarding the next sentence about being "debated" – perhaps it is more precise to say that the origins of circulations in the Himalaya are still the subject of active research.

Thank you for your suggestion. The sentence has been changed to "……*Therefore, the origins of local circulations in the Himalayas remain an active area of research, and their characteristics and driving mechanisms need more in-depth studies……*" in the revised manuscript.

L123. Since the station is in a valley, which profoundly affects the winds, one should be more precise, e.g.., what is being documented is 'the winds in a valley on the north side of Mt. Everest, 30 km to the north of the peak.' Further, this provides insight into the local circulation there. ('on the north slope of Everest' implies being ON the slope, which would imply a different circulation than one in a valley on the north side.). Which is also affected by other nearby terrain.

Thank you for your comments and suggestions. We agree with you. We have emphasized that the data was collected "*in the Rongbuk Valley on the northern side of Mt. Everest*" and the study provides insights into the "*local circulations in the Rongbuk Valley on the north side of Mt. Everest*". Please see lines 122 to 125 in the revised manuscript.

L137. Interaction between the atmosphere, terrain, and surface cover?

"*Land–atmosphere interactions*" refer to the two-way exchanges of water, energy, and momentum between the land surface and the atmosphere above it. Here, "land" refers not only to the surface itself but also to the land cover and underlying surface conditions. This is a command term in this field, generally accepted by the community. We believe it will not cause any misunderstanding. Therefore, we still use the term "land–atmosphere interactions."

L138. Suggest, "The QOMS station is situated at the bottom of the Rongbuk Valley at an elevation of 4276 m above sea level, approximately 30 km north of the summit of Mt. Everest (Figure 1). At the station, the Rongbuk Valley runs in a NNE-SSW direction (approximately 10° to 190°), with a width of about 1.5 kilometers.

(NOTE: 10 to 190 degrees is closer to north-south than to NNE-SSW, but the map suggests something closer to NNE-SSW, which would be more like 22.5 degrees to 202.5 degrees). PLEASE CHECK.

   Thank you for your comments and suggestions. The sentences have been changed to "……*The QOMS station is situated at the bottom of the Rongbuk Valley at an elevation of 4276 m above sea level, approximately 30 km north of the summit of Mt. Everest (Figure 1). At the station, the Rongbuk Valley runs in an NNE-SSW direction (approximately 22.5° to 202.5°), with a width of about 1.5 kilometers……*" in the revised manuscript.

   Moreover, we double checked the valley direction and it is more close to NNE-SSW. Thus, the direction has been changed to 22.5° ~ 202.5°.

L145. Wind direction and speed?

   There was an error in the original manuscript. The are five levels of wind speed (at heights of 1.5, 2, 4, 10, and 20 m) and three levels of wind direction (at heights of 1.5, 10, and 20 m) on the PBL tower.

   The sentence has been changed to "……*The PBL tower is equipped with five levels of air temperature, specific humidity, and wind speed at heights of 1.5, 2, 4, 10, and 20 m, three levels of wind direction at heights of 1.5, 10, and 20 m, and six layers soil temperature and moisture observations buried at depths of 5, 10, 20, 40, 80, 160 cm……*" in the revised manuscript.

L151. Should read 'wind speed and direction' (velocity is a vector). Though, didn't you simply analyze wind speed? You need to be specific about this. People either analyze wind components(u,v,w), horizontal wind components (u,v) or wind speed; and the results obtained for wind speed vs u and v are different. Okay, it's included in the next paragraph.

   Thank you for your comments and suggestions. "Wind velocity and direction" has been revised to "wind speed and direction." We mainly focus on the spectra of horizontal wind speed rather than the individual wind components (u, v, w). This point has been clarified in the title and throughout the manuscript.

Table 1. 'Total' should be 'Total Days'

*"Total" has been changed to "Total Days".*

Section 2.3 okay, although it is important to document how the sonic-anemometer data were processed (corrections for orientation, etc.) and then analyzed.

*Thank you for your comments and suggestions. The sonic-anemometer data were processed using a double rotation orientation correction before the calculation of wind spectra. In the original version, we did not apply a double rotation correction to the sonic-anemometer data as the correction does not affect the calculation of the horizontal wind speed spectrum. As indicated in Figure 1a, the spectra of horizontal wind speed calculated from the unrotated (solid lines) and double-rotated (dashed lines) sonic-anemometer data are overlap with each other. However, the double rotation correction does affect the calculation of the wind component (u, v, w) spectra, as clearly shown in Figure 1b~c.*

*Following the reviewer's suggestion, we applied a double rotation correction to all sonic-anemometer data before computing the spectra. The data processing methods were also introduced. Please see lines 179 to 184 at the end of the first paragraph of Section 2.3 in the revised manuscript: "……Before the calculation of wind speed spectra, the 10-Hz 3D sonic wind data were processed using a double coordinate rotation. In this procedure, the wind components were first rotated around the vertical axis to align the streamwise direction with the mean wind and then rotated around the lateral axis to remove the mean vertical velocity. The main purpose of this treatment is to correct for sensor tilt and local terrain effects, ensuring that the horizontal and vertical wind components are physically consistent and suitable for turbulence and spectral analyses……"*

[Figure]

Figure 1: Comparison of average summer and winter daily spectra of (a) horizontal wind speed, (b) u-component, (c) v-component, and (d) w-component calculated from unrotated (solid lines) and double-rotated (dashed lines) 10-Hz sonic anemometer data collected in 2015 and 2016.

Figure 2. I would have expected the winds going downslope from Everest would provide the dominant direction, which would be from the SSW. But I am not sure what the terrain looks like to the north of the site. So – when you write 'northeasterly to northerly' do you mean 'toward the northeast to north'? (This is one of the frustrating ambiguities of the English language!).

I just looked up "wind rose" on the Internet, which indicates that wind roses indicate the direction from which the wind is blowing, so I am guessing I should interpret the wind is mostly being from the NNE. If so, I would recommend more explicitly writing "north to northeast winds" or even more specifically, "winds from the north to northeast."

Thank you for your comments and suggestions. According to the definition of wind direction, it refers to the direction from which the wind originates. For example, a northeasterly wind means that the wind is blowing from the northeast. To avoid ambiguity, we have revised all descriptions of wind direction in the manuscript. For example, a northeasterly wind is now expressed as "wind from the northeast".

Please see lines 213 to 223 in the revised manuscript: "…… *In spring, winds from the SE to SW (44.1%) and from the NNE (10.5%) are dominate (Figure 3a). Wind speeds from the north are generally less than 6.0 m/s, while a significant number of wind speeds the from south exceed 8.0 m/s. In summer (Figure 3b), the winds primarily blow from NNE (24.1%) and SSE (16.4%) directions, and the wind speed from the SSE is significantly larger than the NNE wind speed. The percentage of wind from other directions is very low. In autumn (Figure 3c), winds from the SSE to SSW (36.8%) and from the NNE (15.1%) are dominate. In winter (Figure 3d), winds from the SW (17.4%) are predominant, followed by winds from the SSW (15.5%), south (13.2%), and NNE (11.0%). Wind speeds from SW are also significantly larger than those from other directions. In general, prevailing winds at the QOMS station are along the valley, blowing from SSE to SW in the spring, autumn, and winter, and from NNE in summer, with higher wind speeds coming from the southern directions. Moreover, wind speeds in winter and spring are larger than those in summer and autumn. Interestingly, winds from the west are almost absent in summer. However, the percentage of winds from west and their speeds both increase gradually from autumn to spring ……*"

Assuming that the dominant wind directions are from the north to northeast, it appears that nearby terrain or terrain to the north has a strong influence as well.

We agree that the surrounding terrain to the north also influence wind speed and direction observed at the QOMS station. To better interpret the results, a zoom in terrain map around the QOMS station was added to Figure 1 in the revised manuscript, which clearly shows obstacles surrounding the station.

The following sentences, which refer to north winds, suggest that most the winds are from the north-northeast, but I think it would be better to avoid use of 'easterly', etc.

Thank you for your comments and suggestions. We have revised the text and replaced "north winds" to "winds from NNE", which is more precise. A clearly dominant wind direction from the NNE is observed in all seasons, especially in summer.

On a related matter, since you are presenting a complete wind rose, it would be useful to have a topographic map with the station at the center. Could be a simplified version. (see other comments about this)

Thank you for your comments and suggestions. We have added a topographic map covering an area of $10 \times 10$ km$^2$ with the QOMS station at the center. Please see Fig 1b in the revised manuscript.

L213-L215. Suggest, for more precision and less ambiguity.

In general, prevailing winds are along the valley, with directions from the SSE to SW in the spring, autumn and winter; and from the NNE in summer; with higher speeds from the southern directions.

    *Thank you for your comments and suggestions. The sentence has been changed to "……In general, prevailing winds at the QOMS station are along the valley, blowing from SSE to SW in the spring, autumn, and winter, and from NNE in summer, with higher wind speeds coming from the southern directions……" in the revised manuscript.*

L218-220. This sentence makes no sense. Why would downward transport of zonal wind (from the west) increase the winds from the south? Please check and correct.

    *We were citing Sun et al. (2018)'s conclusion here, and we also don't quite understand why the downward transfer of westerly momentum would affect the winds from the south. We have deleted this part and revised the sentence to "……Recently, Sun et al. (2018) revealed that the strong SSE winds in Fig. 3 are driven by up-valley winds from the Arun Valley east of Mt. Everest migrating into the Rongbuk Valley where the QOMS station is located……".*

L220-221. For clarification, suggest (If this is correct)

The strong SSW to SW winds in Fig. 3 are driven by .. (rest of sentence okay)

    *Thank you for your suggestion. That should be the strong SE to SSE winds and the sentence was changed to "……Recently, Sun et al. (2018) revealed that the strong SE to SSE winds in Fig. 3 are driven by up-valley winds from the Arun Valley east of Mt. Everest migrating into the Rongbuk Valley where the QOMS station is located……" in the revised manuscript.*

L245. Insert 'for the days"

    *"for the days" has been inserted.*

L250 The existence of aliasing at the highest frequencies should be announced immediately, in Section 2.3, BEFORE the first figure showing spectra.

I.e., Lines 278-280 be deleted, replaced with a statement in Section 2.3 that the increase in spectral energy at the highest frequency is due to aliasing, which results from sampling a signal at too low a rate. This applies to all the spectra shown. It has nothing to do with white noise.

    *Thank you for your comments and suggestions. We have revised the manuscript*

and added a sentence at the end of Section 2.3 to note the presence of aliasing at the highest frequencies, which is "……*It should be noted that an insufficient sampling rate can artificially increase spectral energy at the high-frequency end. Consequently, in this study, the wind speed spectra derived from the 10-Hz sonic wind data all show elevated spectral density in the high-frequency range (as shown in Figures. 5-7……*". Moreover, the sentence indicating this issue in the original manuscript in Section 3.2 has been deleted.

L258. "costal" should be 'coastal'.

Changed.

L260. How about simply contrasting "over land" vs "over the ocean." And, it would be more precise to say the average diurnal peak is smaller over the ocean, rather than 'missing.' You could check whether the Uemaya paper shows a weak diurnal cycle, so that could be cited. A diurnal peak over land is common knowledge, so no citation would be necessary,

Thank you for your comment and suggestion. We have revised the wording and phrasing. We have replaced "missing" with "weaker", and also have deleted the citation for the diurnal peak over land. Please see the sentences in the revised manuscript "…… *The diurnal peak over land is the most pronounced, whereas the diurnal peak over the ocean is comparatively weaker (Ueyama and Deser, 2008). The diurnal peak at the QOMS station is even more significant than previous findings, the power density of the peak is larger than neighbor power densities by more than one order of magnitude ……*"

Regarding the 12-h peak in a mountain area, it would be useful to cite the presence of thermally-driven upslope and downslope winds, citing the book by Whiteman. (I think 'upslope and downslope' winds is sufficiently broad to include upvalley and downvalley winds as well). You could use the adjective "common" or "frequently seen" to indicate that they would not always be observed, since well-defined thermally-driven circulations are most probable in fair-weather, weak-synoptic wind, situations.

Thank you for your comment and suggestion. We have revised the wording and phrasing, and used "frequently" to indicate the 12 $hr^{-1}$ peak would not always be observed. We also cited the book by David Whiteman for the 12 $hr^{-1}$ peak in a mountain area. Please see the sentences in the revised manuscript "…… *The 12 $hr^{-1}$ peak in the wind speed spectrum is frequently expected in mountain area (Whiteman, 2000). The thermally driven diurnal cycle of upslope and downslope winds results in a 12 $hr^{-1}$ cycle in the wind speed. The mountain-valley wind circulation and the*

*associated 12 hr$^{-1}$ peak are pronounced in fair-weather and weak-synoptic conditions……"*

L264. Since the Kang and Wan papers DOES observe a weak 12-hr peak at the Boulder Atmospheric Observatory, contradicting rather than supporting your statement. I would omit citation of this study here. Also, the measurements are not in a mountain area, but near a mountain area.

Thank you for your comments and suggestions. Following your recommendation, we have removed Kang and Won (2016)'s study.

L269-271. There is missing information here. You cannot explain a 12-hour cycle showing up that well with one maximum and one minimum in wind speed in 12 hours.

Thank you for your comments. There was a wrong explanation in the original manuscript. The 12 hr$^{-1}$ peak in wind spectra is primarily caused by the 24-hour cycle in the upslope and downslope wind circulation, but with a 12-hour cycle in the wind speed. This phenomenon was reported by Sun et al., (2018) in their Fig. 3. We have changed the sentence to "…… *The 12 hr$^{-1}$ peak in this study is mainly due to the diurnal transition between calm winds during the evening and morning (21:00~12:00) and strong winds in the afternoon (13:00~20:00) at the QOMS site, with both the calm-wind and the strong-wind periods having their own wind peaks (Sun et al., 2018)……*" in the revised manuscript.

Reference:

Sun, F., Ma, Y., Hu, Z., Li, M., Tartari, G., Salerno, F., Gerken, T., Bonasoni, P., Cristofanelli, P., and Vuillermoz, E.: Mechanism of daytime strong winds on the northern slopes of Himalayas, near Mount Everest: Observation and simulation, Journal of Applied Meteorology and Climatology, 57, 255-272, https://doi.org/10.1175/JAMC-D-16-0409.1, 2018.

L294. Horizontal wind speed?

Comparing Figs 6 and 7 – In Fig. 6, the frequency-weighted sonic spectral energy for horizontal wind is three orders of magnitude lower than that found from the measurements from the tower at 10 m. Yet in Fig. 7, you the sonic energy of the same order of magnitude as that found at the tower at the low-frequency end!

Similarly, the slope of the sonic-derived spectral energy is positive in Fig. 6 over several decades where it is negative in Fig. 7.

Something is wrong here! Data processing? Even believing sonic data at such low frequencies?

Thank you for your comments and suggestion. We apologize if our description or figures were not sufficiently clear and may have led to some confusion. In fact, i**n Figure 6**, **the four solid curves** show frequency-weighted seasonal spectra of horizontal wind speed (10-min wind time series) measured at 10 m in four different seasons. **The four dashed curves** show daily averaged spectra of horizontal wind speed calculated from the 10-Hz sonic-anemometer data in four different seasons. **The two dotted curves** show daily averaged spectra of vertical wind speed in summer (red dotted) and winter (black dotted) calculated from the 10-Hz sonic-anemometer data. **The red (summer) and black (winter) solid curves in Figure 7** are also the frequency-weighted daily spectra of horizontal wind speed calculated from the 10-Hz sonic-anemometer data.

As shown in Figure 6, the spectra calculated from 10-min wind speed and from 10-Hz sonic-anemometer agree well with each other, indicating good consistency of the two types of observations in the overlapping frequency range. Moreover, as shown in Figure 6 (the dashed curves) and Figure 7 (the two solid curves), the daily averaged spectra of horizontal wind speed are consistent in terms of values and structure varying with frequency.

Regarding the slope of spectral energy, slopes of horizontal wind speed spectra in Figure 6 and Figure 7 are all negative. The slopes of vertical wind speed spectra in Figure 7 are positive (the red and black dotted curves).

L303-4. Figures 3 and 4 show wind SPEED not the west-to-east wind component, so this statement needs correcting (or the figure needs correcting). Also, you should make sure the cited papers in line 229-230 are for the corresponding measurement.

Thank you for your comments and suggestion. Here we would like to emphasize that the high spectral density in winter is due to the stronger wind speed compared with the other seasons. The sentence has been changed to "……*In winter, the dominated winds from south to west at the QOMS station are stronger than in other seasons (Figure 3d and Figure 4a), and the combination of strong winds and frequent synoptic weather events lead to the high spectral density*……" in the revised manuscript.

Regarding the cited papers in line 229-230 of the original manuscript, we have double-checked them and make sure they are for the corresponding measurement.

L310. The mesoscale motions being related to the complex terrain (e.g., mountainvalley flows) and deep convection?

We tend to agree with this viewpoint. In the Mt. Everest region, mesoscale motions are strongly influenced by complex topography and high-altitude thermal forcing. Pronounced mountain–valley and glacier wind systems develop in response to strong diurnal heating contrasts among valley floors, mountain slopes, and glacier surfaces, generating organized wind variability on hourly to sub-daily time scales. Moreover, convective activates (not all of them are deep convections) are very frequent in summer at the north side of the Himalayas and produce mesoscale wind disturbances. These terrain- and convection-related processes contribute substantial energy to the mesoscale portion of the wind spectrum (in the frequency range 1 day$^{-1}$ $\leq f \leq$ 1 hr$^{-1}$) and are distinct from small-scale turbulent motions, thereby narrowing or obscuring the classical inertial subrange.

L313. But the basic physics of the Kolmogorov frequency range assumes the smaller eddies simply get their energy from the larger eddies. In this frequency range, the motions are driven by buoyancy associated with nonuniform heating (for complex-terrain flows) and release of latent heat (deep convection). So, it's coincidence. The curious thing is the lack of a -2/3 slope at the higher frequencies, where it's expected (even though there is some energy introduced at higher frequencies).

Thank you for your comments and suggestion. We agree with you that the -2/3 slope in the mesoscale frequency range (1 day$^{-1}$ $\leq$ f $\leq$ 1 hr$^{-1}$) is not really the Kolmogorov frequency range and is just coincidence. We have revised the sentence and deleted the statement "*which seems to be in the Kolmogorov inertial subrange*".

Regarding the -2/3 slope at the higher frequencies, we believe that it does not disappear, rather it becomes very narrow at the QOMS site. As shown in Figure 5 and Figure 6, the frequency-weighted wind speed spectra ($fS(f)$) fall off as -2/3 in the frequency range from about $1\times10^{-1}$ to $2\times10^{0}$ Hz. This frequency range is identified as the Kolmogorov inertial subrange. At the QOMS site, the Kolmogorov inertial subrange is considerably narrower than that typically observed over flat terrain. This is likely due to the strong topographic constraints, which limit the integral length scale and introduce non-local energy inputs through terrain-induced flows. Thus, local energy cascades in complex mountainous terrain are disrupted and turbulence maintains anisotropy down to relatively small scales.

L323. Simply because eddies pass by the sensor faster?

Eddies pass by the sensor more quickly under strong wind conditions, which is one reason. Moreover, the fluctuation is stronger when wind speed is high. They both result in higher spectral densities in strong wind conditions.

L331-332. I question whether the spectra are reliable at these frequencies, due to a combination of aliasing and a significant departure from the expected -2/3 slope.

Thank you for your comments and suggestion. We double checked the data and applied the double rotation correction to the sonic-anemometer data. We believe the data and the data processing have no issues. However, at these high frequencies, instrumental noises are important and could impact on the spectra. We have added one sentence to clarify this issue in the revised manuscript: "……*However, at these high frequencies, increasing amounts of instrumental noise are introduced, leading to greater uncertainty in the spectra......*"

L335 – L339. The existence of a spectral gap is not necessarily a matter of debate; rather the existence of such a gap varies with the situation (and sometimes the method of sampling).

Thank you for your comments and suggestions. To clarify, the sentence has been revised to "……*Current understanding of the interactions between 2D and 3D turbulences in this transition range remains limited. Although it is recognized that the appearance of spectral gaps depends on site-specific conditions and even data sampling methods, the physical mechanisms governing their intensity variations are still not well understood……*"

Figure 7. Lots of problems here! (1) Why is the low-frequency sonic-derived spectral energy three orders of magnitude here higher than in Fig. 6? (2) Moreover, even the slope of the spectral energy as a function of frequency is of opposite sign between 10-3 and 10-1 Hzm=. (3). Eq (1) is arbitrary and contributes nothing, so fitting the low frequencies to match the data has no meaning. (4) Thus, you need only one thick line, to indicate slope – the solid blue line. The red line isn't needed, unless you want to state that the low frequencies don't look like 2-D turbulence. ((1) and (2) noted earlier)

Thank you for your comments and suggestions. We have addressed all your concerns regarding to Figure 7 and revised the manuscript accordingly.

(1) We apologize if our description or figures were not sufficiently clear and may have led to some confusion. We double checked Figure 6 and Figure 7 and found that the sonic-derived spectral energy in the low-frequency range in Figure 6 and Figure 7 is comparable. The frequency-weighted spectral energies ($fS(f)$) are in the range from $2.0 \times 10^{-1}$ to $2.0 \times 10^{0}$ ($m^2 s^{-2}$) both for Figure 6 and Figure 7.

(2) The slopes of the weighted summer and winter daily spectral energy of horizontal wind speed also have the same sign between the frequency range from $10^{-3}$

to $10^{-1}$ Hz, the dashed red and black curves in Figure 6 and solid red and black curves in Figure 7. The reviewer might wrongly take the dotted red and black curves, which are daily vertical wind spectral energy in summer and winter, respectively.

(3) We have deleted all content related to Equation (1) in the revised manuscript.

(4) We have removed the solid red line and kept only the solid blue line showing the reference $f^{2/3}$ spectra.

L349. The spectral gap DID vanish.

We removed "almost" to adopt a more assertive affirmative tone and revised the sentence to "*The spectral gap vanished in the summer spectrum.*"

L350, surpasses that of summer, as noted earlier. Can remove the text after "summer" as this was already discussed in detail.

(I am ignoring discussion of the dubious eq (1))

Thank you for your comments and suggestions. We have deleted the text after "summer".

L376-L380, figure 8. Something is very weird about this plot. It suggests that turbulence intensity is higher with weaker wind, quite the opposite of what was discussed earlier. I suggest that this section either be corrected or deleted altogether.

Thank you for your comments and suggestions. We have made responses to your general comments and just repeated our responses here.

We apologize if our description was not sufficiently clear and may have led to some confusion. In fact, in the atmospheric boundary layer, turbulence intensity ($TI = \sqrt{\sigma_u^2 + \sigma_v^2}/U$) generally decreases with increasing mean wind speed. This is because as wind speed increase, although turbulent fluctuations ($\sigma_u$, $\sigma_u$) may still intensify, their relative contribution decreases, leading to a decreasing trend of TI with wind speed. This phenomenon has been discussed in Panofsky and Dutton (1984) and Stull (1988), and has been observed by Ernst and Seume (2012) and Kang and Won (2016).

In Figure 8, the black line shows the mean value of TI in winter 2016, while the red line shows the TI in summer. Here we would like to show that the TI in winter is higher than that in summer, as the wind speed is stronger in winter than in summer (Figure 3 and Figure 4). Moreover, the spectral density in winter is also higher than in summer in the microscale frequency range (Figure 6). Thus, wind speed, spectral density, and turbulence intensity discussed in the study are consistent with each other.

References:

1. Ernst, B. and Seume, J. R.: Investigation of Site-Specific Wind Field Parameters and Their Effect on Loads of Offshore Wind Turbines, Energies, 5, 3835-3855, 2012.
2. Kang, S.-L. and Won, H.: Spectral structure of 5 year time series of horizontal wind speed at the Boulder Atmospheric Observatory, Journal of Geophysical Research: Atmospheres, 121, 11,911-946,967, 2016.
3. Panofsky, H. A. and Dutton, J. A.: Atmospheric turbulence: Models and methods for engineering applications, John Wiley & Sons, New York, 1984.
4. Stull, R. B.: An introduction to boundary layer meteorology, Kluwer Academic Publishers, Dordrecht, 1988.

L405-406. For the 12-h spectral peak, should check nighttime winds, too, as noted before.

Thank you for your comments. There was an incorrect explanation in the original manuscript. We have revised the description in Section 2, stating that "……*the 12 h$^{-1}$ peak in this study is mainly due to the diurnal transition between calm winds during the evening and morning (21:00–12:00) and strong winds in the afternoon (13:00–20:00) at the QOMS site, with both the calm-wind and strong-wind periods exhibiting their own wind peaks (Sun et al., 2018) ……*" We have also revised the corresponding statement in the Summary and Conclusions section to read: "……*The spectral peak at 12 h$^{-1}$ is most likely related to the diurnal local circulations in the valley where the QOMS station is located ……*"

414. There is no spectral gap in summer.

Thank you for your comment. The sentence has been revised to "……*It is the most obvious in the winter season while vanished in summer……*"